# Integrated transcriptome and proteome analysis reveals posttranscriptional regulation of ribosomal genes in human brain organoids

Jaydeep Sidhaye[1†], Philipp Trepte[1†], Natalie Sepke[1], Maria Novatchkova[1], Michael Schutzbier[2], Gerhard Dürnberger[2], Karl Mechtler[1], Jürgen A Knoblich[1,3]*

[1]Institute of Molecular Biotechnology of the Austrian Academy of Sciences (IMBA), Vienna BioCenter (VBC), Vienna, Austria; [2]Gregor Mendel Institute, Vienna Biocenter, Vienna, Austria; [3]Department of Neurology, Medical University of Vienna, Vienna, Austria

*For correspondence:
juergen.knoblich@imba.oeaw.ac.at

[†]These authors contributed equally to this work

**Abstract** During development of the human cerebral cortex, multipotent neural progenitors generate excitatory neurons and glial cells. Investigations of the transcriptome and epigenome have revealed important gene regulatory networks underlying this crucial developmental event. However, the posttranscriptional control of gene expression and protein abundance during human corticogenesis remains poorly understood. We addressed this issue by using human telencephalic brain organoids grown using a dual reporter cell line to isolate neural progenitors and neurons and performed cell class and developmental stage-specific transcriptome and proteome analysis. Integrating the two datasets revealed modules of gene expression during human corticogenesis. Investigation of one such module uncovered mTOR-mediated regulation of translation of the 5'TOP element-enriched translation machinery in early progenitor cells. We show that in early progenitors partial inhibition of the translation of ribosomal genes prevents precocious translation of differentiation markers. Overall, our multiomics approach proposes novel posttranscriptional regulatory mechanisms crucial for the fidelity of cortical development.

## Editor's evaluation

This fundamental work integrates transcriptome and proteome analysis across human neurogenesis, uncovering posttranscriptional regulatory mechanisms for a specific gene module enriched in ribosomal genes. The evidence supporting the conclusions is compelling, exploiting a range of targeted human pluripotent stem cell lines for brain organoid generation. The work will be of broad interest to developmental and neurobiologist.

## Introduction

The development of the human cerebral cortex is a highly elaborate process occurring over several months. During this process multipotent neural progenitors give rise initially to excitatory neurons of the different layers of the cerebral cortex and later to glial cells. Studies using mouse model systems, human fetal samples, as well as brain organoids have revealed elaborate spatiotemporal gene regulatory networks that orchestrate mammalian corticogenesis (*Cadwell et al., 2019*; *Greig et al., 2013*; *Shibata et al., 2015*; *Vaid and Huttner, 2020*). However, an emerging theme from mouse corticogenesis studies highlights the additional role of posttranscriptional gene regulatory mechanisms, such

as alternative splicing and translational repression, in determining progenitor cell fate and neuronal migration (*Hoye and Silver, 2021*; *Lennox et al., 2017*; *Zahr et al., 2019*). Whether similar mechanisms play a role in human neurodevelopment remains elusive. Hence, investigating the posttranscriptional control of gene expression and protein abundance is crucial to comprehensively understand gene regulation during human neurodevelopment.

Transcriptome analyses have placed a lot of importance on transcript abundance, however, the relationship between transcript and protein abundance remains elusive. While previous studies have investigated posttranscriptional regulation of specific genes or the impact of loss of key RNA-binding proteins (RBPs) (*Hoye and Silver, 2021*), proteome-scale analyses of gene regulation during human corticogenesis have been lacking. Currently available proteome data from hiPSC-derived neural progenitors and neurons (*Djuric et al., 2017*; *Varderidou-Minasian et al., 2020*), cerebral organoids (*McClure-Begley et al., 2018*; *Melliou et al., 2022*; *Nascimento et al., 2019*), and the fetal brain (*Djuric et al., 2017*; *Kim et al., 2014*; *Melliou et al., 2022*) are an important step toward establishing a human neurodevelopmental proteomic repertoire. However, these studies suffer from some of the following limitations: (a) due to the inherent diversity of cell types present during corticogenesis, bulk tissue data do not provide cell type-specific information, (b) cell sorting strategies result in datasets that suffer from low number of successfully detected proteins, and (c) importantly, a direct comparison of the transcript and protein expression to understand posttranscriptional gene regulation has been missing.

We addressed these issues by using a dual reporter cell line to separate progenitors and neurons from the human brain organoid tissue and performed cell class- and developmental stage-specific transcriptome and proteome analysis. We integrated the two datasets to identify gene expression modules active during neural progenitor proliferation and subsequent neurogenesis in the developing cortical tissue. This dataset is available as a resource for the community in the form of a Shiny app (https://organoid.multiomics.vbc.ac.at). We followed up on one gene expression module in detail and found that the genes of the core translational machinery, containing the '5' terminal oligopyrimidine' (5'TOP) motif in their 5'UTR, are coregulated posttranscriptionally during neurodevelopment. We show that in contrast to neurons, translation of 5'TOP transcripts is partially inhibited in early progenitors, resulting in discordant RNA-protein relationship between the two cell classes. This regulation of the 5'TOP element-enriched translational machinery is due to lower mTOR activity in early progenitors and is crucial for the fidelity of cortical development.

## Results

### RNA-protein multiomics of neural progenitors and neurons isolated from telencephalic organoids

To investigate progenitor and neuron-specific gene regulatory programs, we generated a dual reporter H9 human embryonic stem cell (hESC) line that enables sorting of neural progenitors and neurons (*Figure 1A*). To label neural progenitors, we replaced the stop codon of one allele of the pan-neural progenitor marker gene *SOX2* with P2A-EGFP (*Figure 1—figure supplement 1A*). Tagging the endogenous gene would recapitulate the endogenous gene regulatory programs with minimal interference. On the other hand, to label neurons, we inserted dTomato driven by human Synapsin1 (hSYN1) promoter in the AAVS1 safe harbor locus (*Figure 1—figure supplement 1B*). Using this dual reporter line, we generated brain organoids enriched in dorsal telencephalic tissue using a previously published organoid culture protocol (*Esk et al., 2020*; *Figure 1—figure supplement 1C*) to recapitulate the key stages of corticogenesis, when multipotent progenitors give rise to diverse excitatory neurons of distinct layer identities. Immunostaining for EGFP and dTomato confirmed broad expression of the fluorophores in the progenitor and neuronal zones, respectively (*Figure 1B*, *Figure 1—figure supplement 1D*). Analysis of sparsely labeled organoids set up using 5% dual reporter hESCs and 95% non-reporter, wild-type hESCs allowed us to dissect the expression of the markers at cellular level. First, it confirmed expression of EGFP and dTomato in the progenitor-rich ventricular zone (VZ) (SOX2), and the surrounding neuron-rich region (NeuN), respectively (*Figure 1—figure supplement 2A–C*). It was observed that in accordance with their cytoplasmic localization, both fluorescent proteins marked the entire cell body and therefore also the long cellular processes characteristic of radial glia and neurons. A few cells expressed both EGFP and dTomato probably indicative of a

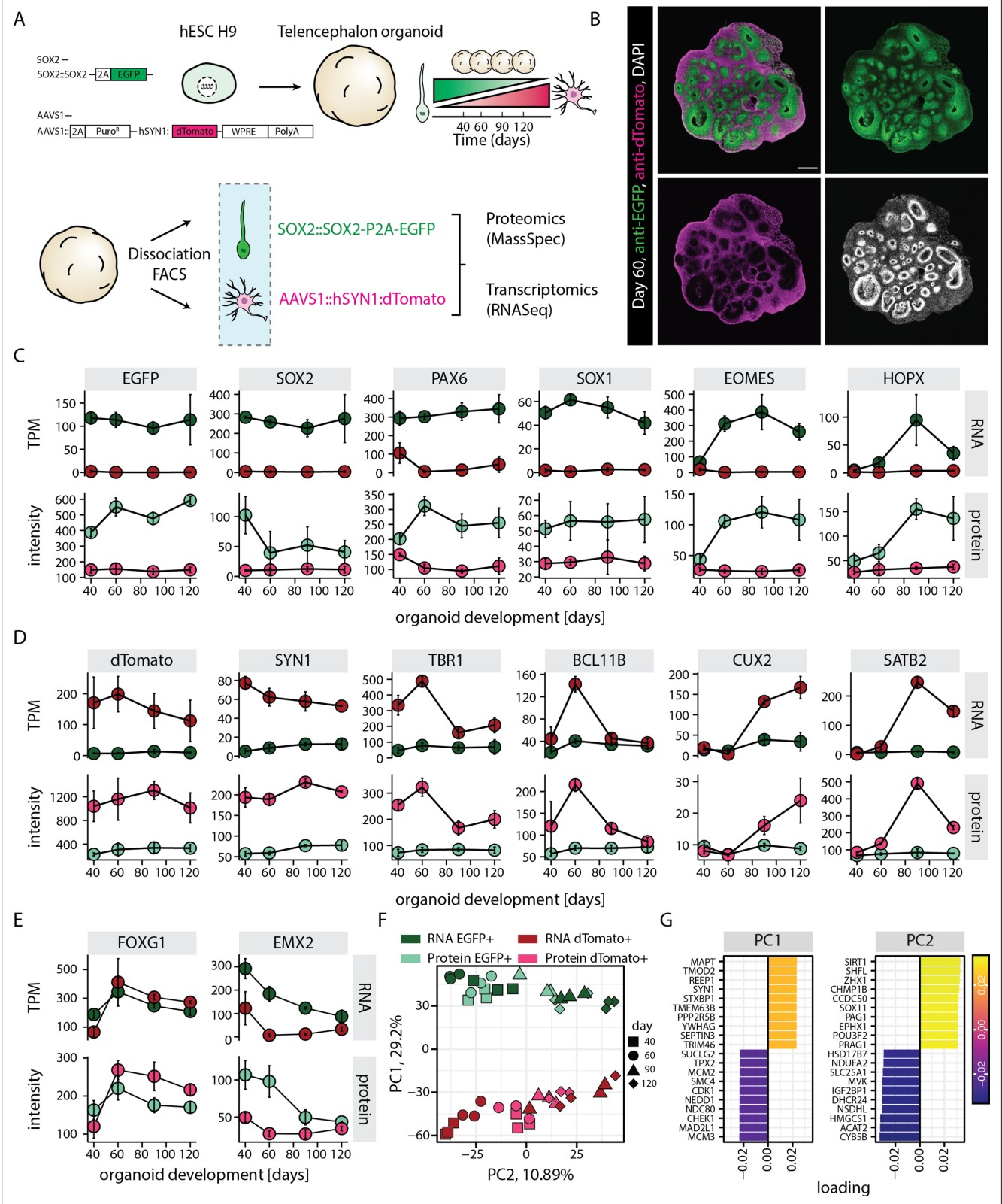

**Figure 1.** RNA-protein multiomics of progenitors and neurons in human brain organoids. (**A**) Schematic representation of the experimental design. A dual reporter line was generated in H9 human embryonic stem cell (hESC) background. Cells from brain organoids grown using this reporter line were sorted and collected by fluorescent activated cell sorting (FACS) in EGFP and dTomato positive fractions. Gene expression of the sorted cell populations were analyzed by RNA-seq and Mass-spec. (**B**) Confocal scan of a dual reporter organoid cryosection (day 60) stained with anti-EGFP, anti-dTomato

*Figure 1 continued on next page*

*Figure 1 continued*

antibodies, and DAPI. Scale bar = 500 μm. (**C,D,E**) Plots showing RNA (top) and protein (bottom) abundance of key progenitor-specific markers (**C**), neuronal markers (**D**), and dorsal telencephalic fate markers (**E**) in *SOX2*::EGFP+ (green) and *hSYN1*::dTomato+ positive (red) cells at different stages of organoid development. RNA abundance is measured as transcript per million (TPM). (**F,G**) Principal component analysis of z-score normalized integrated transcriptome and proteome of progenitors (EGFP+, green) and neurons (dTomato+, red) at different stages of organoid development with (**G**) loading scores for PC1 and PC2 top contributing genes.

The online version of this article includes the following source data and figure supplement(s) for figure 1:

**Source data 1.** RNA-seq gene expression results.

**Source data 2.** Mass spectrometry protein expression results.

**Source data 3.** Combined RNA and protein gene expression results and module assignment of genes.

**Figure supplement 1.** Generation of a dual reporter human embryonic stem cell (hESC) line to differentially label neural progenitors and neurons.

**Figure supplement 1—source data 1.** Original unlabelled files (.scn and .tif) of gel picture for PCR confirming insertion of SOX2-P2A-EGFP casette (see *Figure 1—figure supplement 1A*) and Puro-hSyn1::dTomato-WPRE-pA casette (see *Figure 1—figure supplement 2Bii*).

**Figure supplement 1—source data 2.** Labelled file of gel picture for PCR confirming insertion of SOX2-P2A-EGFP casette (see *Figure 1—figure supplement 1A*) and Puro-hSyn1::dTomato-WPRE-pA casette (see *Figure 1—figure supplement 2Bii*).

**Figure supplement 1—source data 3.** Original unlabelled files (.scn and .tif) of gel picture for PCR confirming insertion of Puro-hSyn1::dTomato-WPRE-pA casette (see *Figure 1—figure supplement 2Bi*).

**Figure supplement 1—source data 4.** Labelled file of gel picture for PCR confirming insertion of Puro-hSyn1::dTomato-WPRE-pA casette (see *Figure 1—figure supplement 2Bi*).

**Figure supplement 1—source data 5.** Original unlabelled files (.scn and .tif) of gel picture for PCR confirming insertion in AAVS1 locus (see *Figure 1—figure supplement 2Biii*).

**Figure supplement 1—source data 6.** Labelled file of gel picture for PCR confirming insertion in AAVS1 locus (see *Figure 1—figure supplement 2Biii*).

**Figure supplement 2.** Characterization of the dual reporter using sparse labeling strategy.

**Figure supplement 3.** Fluorescent activated cell sorting (FACS) from the dual reporter organoids.

**Figure supplement 4.** Principal component analysis (PCA) of transcriptome and proteome of progenitors and neurons sorted from dual reporter organoids.

**Figure supplement 5.** Characterization of cell populations fluorescent activated cell sorted from the dual reporter organoids.

**Figure supplement 6.** Comparison of proteome of intact organoids and cells sorted using the dual reporter strategy.

**Figure supplement 7.** Integration and clustering of the RNA-protein multiomics data.

transition state (*Figure 1—figure supplement 2D*). Furthermore, some EGFP positive cells were also located outside the typical VZ and stained positive for EOMES, a marker for intermediate progenitors (*Figure 1—figure supplement 2D*). Thus, the dual reporter strategy exhibited the expected neuro-developmental expression for progenitors and neurons.

To study progenitor and neuron-specific gene regulatory programs, we aimed to perform whole transcriptome and proteome analysis of the respective cell classes. Dissociation of the organoids followed by fluorescence activated cell sorting (FACS) confirmed the occurrence of EGFP positive (EGFP+) and dTomato positive (dTomato+) cell populations across different stages of organoid development (days 40, 60, 90, and 120) (*Figure 1—figure supplement 3A*). Additionally, an EGFP/dTomato double positive cell population was detected (*Figure 1—figure supplement 3A*). RNA-seq-based whole transcriptome analysis of these sorted cell populations and subsequent principal component analysis (PCA) revealed that the double positive population at days 40 and 60 matched a transition stage between progenitors and neurons, while at days 90 and 120 it was more similar to EGFP+ cells (*Figure 1—figure supplement 3B*). However, we focused on cells exclusively positive for single fluorophores as stringent criteria for bona fide progenitors (*SOX2*::EGFP+) and neurons (*hSYN1*::dTomato+). PCA showed that *SOX2*::EGFP+ cells and *hSYN1*::dTomato+ cells clustered separately and showed transcriptomic signatures of progenitors and neurons, respectively (PC1, *Figure 1—figure supplement 4A and B*). These samples further separated according to the organoid developmental stages with little batch effect (PC2, *Figure 1—figure supplement 4A and B*).

For proteome analysis, proteins from the cells sorted from the same organoid culture batches and time points as used for RNA-seq were tandem mass tag (TMT) labeled and combined for each batch separately. In total, 6740 proteins were detected by mass spectrometry of the sorted cell populations.

Initial analysis confirmed that similar to the transcriptome, the proteome of sorted neurons (*hSYN1*::d-Tomato+) and progenitor cells (*SOX2*::EGFP+) segregated according to the cell class (*Figure 1—figure supplement 4C and D*). Segregation by age was less striking potential due to technical limitations reported for TMT-labeling experiments, such as ratio compression (*Savitski et al., 2013*) and channel cross-talk (*Searle and Yergey, 2020*).

Transcriptome and proteome analysis (*Figure 1C–E*) confirmed that EGFP and dTomato expression matched the expression of endogenous *SOX2* and *SYN1*, thus highlighting the suitability of our dual reporter strategy (*Figure 1C and D*). Furthermore, this analysis revealed that the *SOX2*::EGFP+ population encompassed all different classes of cortical progenitors including ventricular radial glia (expression of *SOX1*, *PAX6*), intermediate progenitors (*EOMES*), as well as outer radial glia (*HOPX*). The expression patterns of these markers followed the expected temporal trajectories of cortical development. While the markers of ventricular radial glia were expressed at all stages, markers of intermediate progenitors and outer radial glia showed enrichment at days 60 and 90, respectively (*Figure 1C*). Similarly, the *hSYN1*::dTomato+ population included excitatory neurons of all different cortical layers and followed the expected temporal expression patterns (*Figure 1D*), with deep-layer markers (*TBR1, BCL11B*) being expressed before upper-layer markers (SATB2, CUX2). The enrichment of dorsal telencephalic fate was confirmed by the expression of above stated markers of cortical progenitors and neurons along with high expression of FOXG1 and EMX2 (*Figure 1E*) and undetectable levels of ventral markers (NKX2.1, LHX6) (TPM values below 1). In line with the PCA, the excluded double positive population at days 40 and 60 showed expression of progenitor and neuronal markers at levels intermediate to those in *SOX2*::EGFP+ population and *hSYN1*::dTomato+ population. This is indicative of a transition state between progenitors and neurons (*Figure 1—figure supplement 5A*). However, at days 90 and 120, the double positive population showed mixed marker expression including high expression of general progenitor and outer radial glia markers, enrichment of gliogenic markers, as well as moderate expression of neuronal markers (*Figure 1—figure supplement 5A*). These observations suggest that the double positive population does not represent a particular cell class but rather cellular transition states and mixtures of different cell types. It further supported our decision to focus on cells exclusively positive for single fluorophores.

Finally, we asked if the organoid dissociation and cell sorting strategy resulted in the loss of specific proteins and cell types. Therefore, we performed a proteome analysis of intact organoids pooling the four time points and compared the protein expression to a pool containing all sorted cells (*Figure 1—figure supplement 6A*). We observed that mainly extracellular matrix (laminins, collagens) and cell adhesion proteins (e.g. AGRN, VCAN), as well as markers for astrocytes (GFAP) and choroid plexus (TTR) were lost upon tissue dissociation and dual reporter-based FACS (*Figure 1—figure supplement 6B*). Overall, we generated a time-resolved transcriptome and proteome dataset of neural progenitors (*SOX2*::EGFP+) and neurons (*hSYN1*::dTomato+) sorted from telencephalic organoids (*Figure 1—source data 1*, *Figure 1—source data 2*).

## Integrated transcriptome and proteome analysis reveals gene expression modules active during neurodevelopment

Next, we examined the general degree of correlation between mRNA and protein levels. We detected mean TPM values greater than 0 and at least one peptide for 6714 genes. Similar to a previous report (*Schwanhäusser et al., 2011*), we observed only a limited correlation between absolute mRNA and protein levels, which seemed comparable between cell classes and developmental time points (*Figure 1—figure supplement 7A*). Given the very different measurement units of RNA-seq and Mass-spec, a PCA on the combined datasets revealed that almost 80% of the variation was explained by the omics method (PC1, *Figure 1—figure supplement 7B*) and not by cell class or developmental time point highlighting the importance of data normalization. Furthermore, to identify groups of genes that follow similar relative gene expression profiles at RNA and protein level in progenitors and neurons over time, we combined the two datasets upon rescaling using gene-by-gene z-score normalization (*Figure 1—source data 3*). PCA of the combined datasets confirmed that the scaled data grouped by cell classes and developmental age and not by the omics method (*Figure 1F and G*). Using the z-score scaled dataset we first fit a global regression model to identify genes differentially expressed between the two cell classes on RNA and/or protein level. In a second regression step, genes showing significant temporal profile differences for RNA and protein expression in progenitors and neurons were

identified (*Nueda et al., 2014*). Out of the 6714 genes, we could successfully fit regression models for 5978 genes, 3668 of which showed significant temporal expression changes between cell classes on RNA and/or protein levels. Finally, we performed hierarchical clustering to identify genes with underlying common trends of RNA-protein expression. To estimate an optimal number of biologically meaningful clusters, we used the 'within cluster sums of squares', 'average silhouette', and 'gap statistics' methods (*Figure 1—figure supplement 6C*). In the end, a total of 3368 genes were classified into nine modules (*Figure 2A and B*). The remaining 2310 genes were considered as 'not clustered' and 736 genes for which the first regression fit failed as 'not fit' (*Figure 1—figure supplement 7D and E* and *Figure 1—source data 3*). These genes lacked specific expression patterns and were enriched in house-keeping genes, including core components of the proteasome, spliceosome, and RNA polymerase (*Figure 1—figure supplement 7F*). Thus, we were able to identify modules of genes showing distinct temporal and cell class-specific expression patterns (*Figure 1—source data 3*).

Functional enrichment analysis helped to characterize the members of each module (*Figure 2C*, *Figure 2—figure supplement 1A and B*). Furthermore, the average global trends of relative expression of RNA and proteins helped to postulate potential regulatory mechanisms for each module (*Figure 2B*). Additionally, we examined the absolute expression levels of a few individual member genes, to verify that the global trends are reflected for individual genes without normalization (*Figure 2—figure supplement 2A*). Modules 1, 2, 7, 8, and 9 showed very strong cell class-specific expression patterns matching the sorting strategy for progenitors and neurons (defined here as 'C' modules) (*Figure 2B*). Modules 1 and 2 showed higher relative expression in the progenitors than neurons across all time points and were enriched in DNA replication and cell activation-related genes, respectively (*Figure 2C* and *Figure 2—figure supplement 1A and B*). For module 2, progenitor-enriched expression increased with time, in line with the observation that many outer radial glia markers were members of this cluster. Genes in modules 7, 8, and 9 showed higher expression in neurons than progenitors across all time points. Module 7 was enriched in genes related to mitochondrial respiration (*Figure 2C*, *Figure 2—figure supplement 1A and B*), which is in agreement with the metabolic shift from glycolysis to oxidative phosphorylation observed during neurogenesis (*Iwata and Vanderhaeghen, 2021*). Modules 8 and 9 were enriched in axonal and pre- and postsynaptic genes (*Figure 2C* and *Figure 2—figure supplement 1A and B*). Importantly, in 'C' modules, global relative expression of RNA and protein followed largely similar trends, indicating that the regulation of these genes might occur via cell class-specific transcription or transcript retention and decay.

Interestingly, the global profiles of relative RNA and protein expression did not follow similar trends for other modules, which instead showed a temporal or cell class-specific discrepancy. Modules 4 and 6 showed relative gene expression changing with time ('T' modules) (*Figure 2B*). In module 4, expression decreased with time and the average RNA trends for these genes were similar for progenitors and neurons. However, progenitors expressed relatively higher amounts of protein. Such a trend might be explained by less efficient translation or reduced protein stability of these genes in neurons. This module includes known early fate specification genes such as *FEZF2*, as well as genes related to glycolysis, a process more prominent in progenitors (*Iwata and Vanderhaeghen, 2021*). In contrast, module 6 genes were expressed more at later stages. The average RNA trends of these genes were similar for progenitors and neurons, but protein levels were higher in neurons (*Figure 2B*), suggesting less efficient translation or reduced protein stability in progenitors over neurons. This cluster includes upper-layer fate regulators such as SATB2 and CUX2.

The remaining modules 3 and 5 showed more ambiguous patterns of expression ('A' modules) (*Figure 2B*). Genes from module 3 showed considerably higher transcript levels in progenitors than neurons, although protein levels between the two cell classes were more similar. This observation suggested that translation of these genes is either less efficient in progenitors or enhanced in neurons, or that protein stability is different between the two cell classes. Gene enrichment analysis suggested enrichment of translational machinery proteins including those of ribosomes (*Figure 2C*, *Figure 2—figure supplement 1A and B*). Lastly, module 5 genes showed opposite patterns for RNA and protein relation in progenitors and neurons. The relative RNA levels showed a temporal increase in the two cell classes, with late-stage neurons expressing even higher amounts than corresponding progenitors. Despite this trend, the relative protein levels were higher in progenitors than neurons and with the difference being highest at early stages. This pattern suggested higher translational efficiency or higher protein stability of these genes in progenitors over neurons (*Figure 2B*). This module is

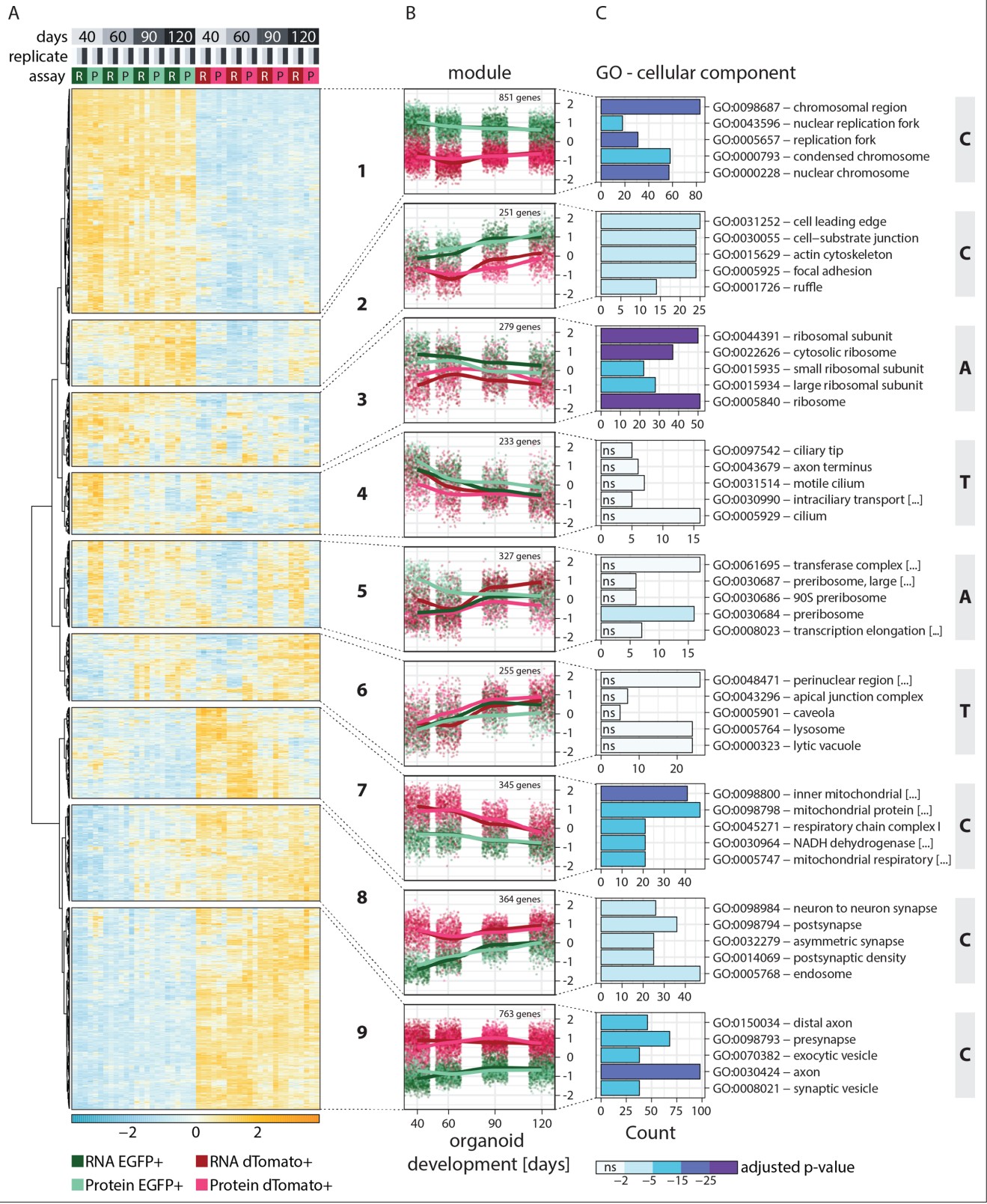

**Figure 2.** Clustering of genes based on relative RNA-protein expression patterns in progenitors and neurons in human brain organoids. (**A**) Heatmap showing clustering results for the nine gene expression modules. Z-score normalized abundance of RNA (**R**) and protein (**P**) in progenitors (EGFP+, green) and neurons (dTomato+, red) at different stages of organoid development. (**B**) Z-score normalized RNA and protein abundance in progenitors (EGFP+, green) and neurons (dTomato+, red) at different stages of organoid development for the nine modules. For each dataset, each dot displays

*Figure 2 continued on next page*

*Figure 2 continued*

the relative abundance of one gene and a trendline was fit through all data points. (**C**) Enrichment analysis for gene ontology (GO) cellular component (CC) for genes in the nine modules. Shown are the top five enriched terms per module and the number of genes (x-axis) belonging to the respective GO term. Bar fill colors show binned adjusted p-values and not significant (ns) terms are indicated. Modules are categorized according to their global expression pattern: cell class-specific expression ('C' modules), temporal expression ('T' modules), and ambiguous expression ('A' modules).

The online version of this article includes the following figure supplement(s) for figure 2:

**Figure supplement 1.** Gene ontology (GO) analysis of gene expression modules.

**Figure supplement 2.** Analysis of gene expression modules and correlation between relative RNA-protein abundance.

enriched in RBPs like DEAD box proteins and proteins important for rRNA processing and ribosome biogenesis (*Figure 2C*, *Figure 2—figure supplement 1A and B*). Finally, we analyzed the correlation of the relative abundance of RNA to protein for each gene expression module (*Figure 2—figure supplement 2B*). Modules that show a high cell class-specific expression ('C' modules) showed a higher correlation than modules with broader expression patterns ('T' and 'A' modules) reflected in the coherency between RNA and protein trends. Thus, by combining RNA and protein expression data, we identified gene modules with cell class-specific and temporal gene expression patterns, as well as modules with seemingly ambiguous expression patterns that show divergent coherency between relative RNA and protein abundance.

## Analysis of module-specific RNA regulatory features highlights the 5'TOP gene module

The emergence of highly distinct expression modules raises the possibility that the genes within each module are regulated transcriptionally and posttranscriptionally by common cell class- or stage-specific mechanisms. The posttranscriptional mechanisms can include regulation of translation as well as stability of the mRNAs and proteins, either uniformly or partially through compartmentalization. As transcript features of the 5' and 3'UTR elements are postulated to be a major factor that directly influences RNA stability and translation (*Blair et al., 2017*; *Floor and Doudna, 2016*), we performed a comprehensive transcript feature analysis of the members of the nine modules (*Figure 3A*, *Figure 3—source data 1*, *Figure 3—source data 2*).

First, we analyzed trans-regulatory features that can potentiate binding of RBPs and miRNAs (*Figure 3A*). To this end, we performed module-wise over- and underrepresentation analysis for RBP-binding motifs in the 5' and 3'UTR elements of the member genes (*Figure 3B–C*, *Figure 3—figure supplement 1A–B*, *Figure 3—source data 1*). Modules 1 and 9, whose member genes showed most cell class-specific expression, showcased high numbers of RBP motifs that were significantly enriched or depleted in the 3'UTR (*Figure 3B*). The RBPs identified in our analysis include several RBPs (SMAUG2/SAMD4B, IMP1/IGF2BP1, HUR/ELAVL1, NOVA2, PTBP1, RBFOX1) previously shown to play an important role in neurodevelopment (*Lennox et al., 2017*). Upon comparing over- and under-represented motifs among different modules, we observed that there was a reciprocal relationship between motifs enriched in the progenitor modules 1 and 2 and the neuronal module 9 (*Figure 3C*). These data suggest that the RBP motifs we identified and their associated proteins might play a prominent role in cell class-specific expression to promote progenitor or neuron-specific gene regulation. For instance, we observed that motifs for the RBP RBFOX1, which was shown to promote neuronal expression of synaptic genes (*Lee et al., 2016*), were enriched in the 3'UTRs of module 9 (*Figure 3C*) in line with the enrichment for gene ontology (GO) terms for synaptic genes (*Figure 2C*). As another example, ELAVL1 protein was shown to be important to maintain progenitor state by stabilization of key cell cycle and notch signaling transcripts (*García-Domínguez et al., 2011*). In line with this, motifs for the ELAVL family, especially ELAVL1, were enriched in the 3'UTRs of modules 1 and 2 (*Figure 3C*, *Figure 3—source data 1*) that showed enrichment for GO terms for cell cycle and neural progenitor genes (*Figure 2C*, *Figure 2—figure supplement 1*).

Next, we searched for miRNA-binding sites in the 3'UTR, which are known to repress the expression of transcripts they bind to. Module-wise analysis showed a significantly higher density of miRNA-binding sites for genes of modules 6 and 8, indicating that the expression of some fate regulators and neuronal genes is regulated by miRNAs (*Figure 3—figure supplement 1C*). For example, the upper-layer neuronal fate regulator gene SATB2, a member of module 6 is regulated through the

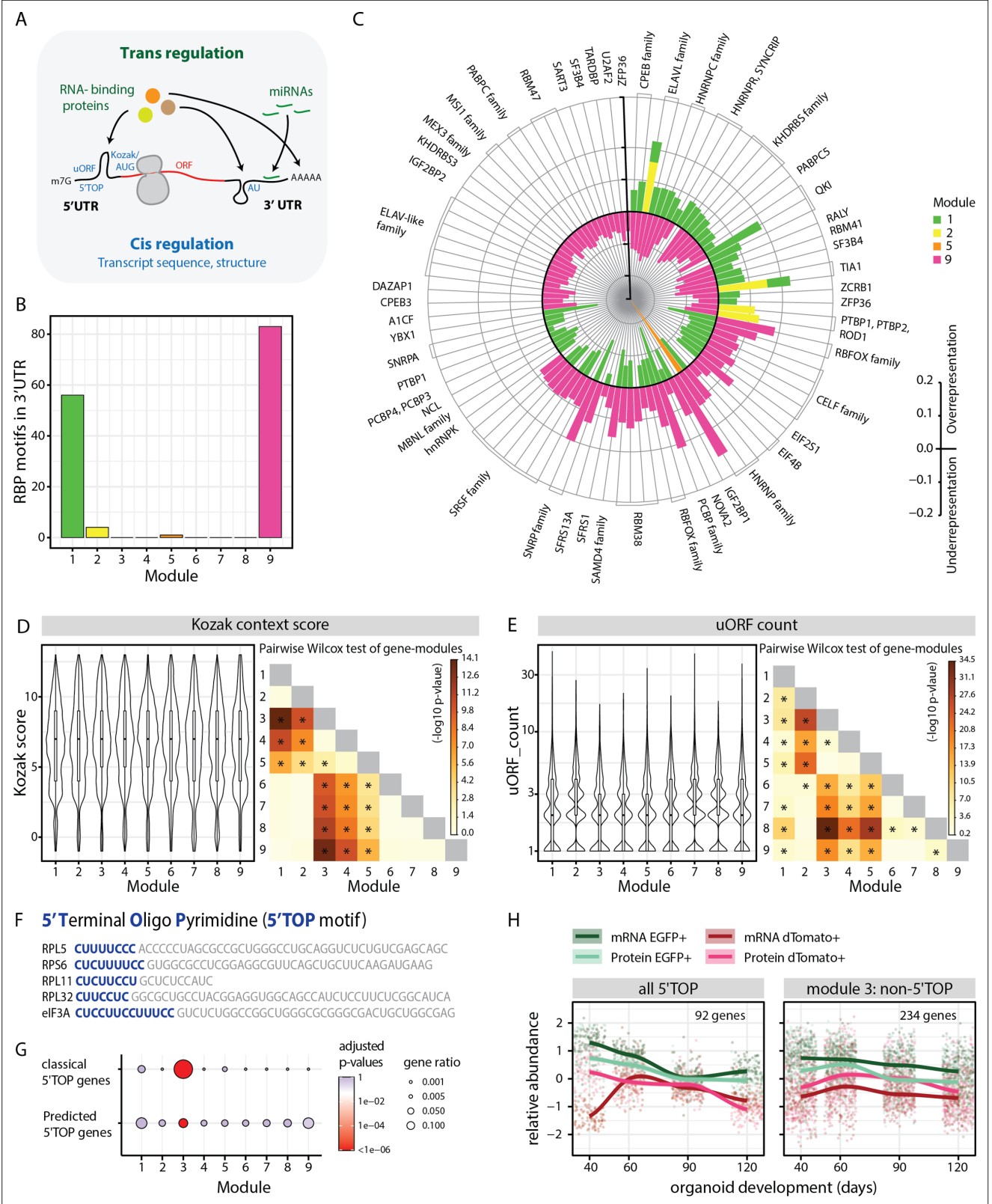

**Figure 3.** Analysis of module-specific RNA regulatory features. (**A**) Schematic showing trans-regulation of mRNA translation by RBPs and miRNAs and cis-regulation by mRNA sequence features and secondary structure. (**B**) Barplot showing number of RBP motifs with significant over/underrepresentation in the 3'UTRs of genes for each module. Significance threshold: adjusted p-value <0.05. (**C**) Circular barplot showing RBP motifs that are significantly over/underrepresented in modules in (**B**). Black circle marks zero position. Positive values outside the black circle indicate overrepresentation. Negative

*Figure 3 continued on next page*

*Figure 3 continued*

values inside the black circle indicate underrepresentation. RBPs for specific motifs are mentioned on the circular axis ticks. Bars are colored according to the module. (**D,E**) Violin plots of module-wise distribution of Kozak context scores (**D**) and number of uORFs (**E**) for transcripts of member genes. Inner boxplots mark median Kozak context scores and interquartile range (IQR). Whiskers extend to 1.5× IQR. The heatmap on the right shows the -log10 p-value of the pairwise Wilcoxon test of gene modules. Significant comparisons are marked by an asterisk (p<0.01). (**F**) 5' terminal sequences of five selected 5' terminal oligopyrimidine (5'TOP) transcripts. Pyrimidine-rich 5'TOP sequences are highlighted in blue. (**G**) Dot plot showing overrepresentation analysis of 5'TOP genes across the nine modules. Significance threshold: adjusted p-value < 0.05. Gene lists refer to classical 5'TOP genes (*Philippe et al., 2020*) and predicted 5'TOP mRNAs (*Yamashita et al., 2008*; *Figure 3—source data 3*). (**H**) Z-score normalized relative abundance of transcripts and proteins in progenitors and neurons of all 5'TOP genes and of module 3 excluding 5'TOP genes.

The online version of this article includes the following source data and figure supplement(s) for figure 3:

**Source data 1.** RNA-binding protein motifs in the 3' and 5'UTRs of module genes.

**Source data 2.** Transcript feature analysis for the modules.

**Source data 3.** Classical and predicted 5'TOP genes.

**Figure supplement 1.** Analysis of module-specific trans- and cis-regulatory features.

binding of miR-541 and miR-92a/b to its 3'UTR thereby preventing its translation (*Martins et al., 2021*). Altogether, these data indicated that trans-regulatory factors play a major role in the regulation of transcripts with cell class-specific expression ('C' modules).

Next, we analyzed cis-regulatory features inherent in the mRNA sequence (*Figure 3A*, *Figure 3—source data 2*). We compared the lengths of coding sequences (CDS) and the 5' and 3'UTRs (*Figure 3—figure supplement 1D–F*). Compared to all other modules, module 3 transcripts were overall shorter in all these features, whereas modules 8 and 9 seemed to have longer 5' and 3'UTRs (*Figure 3—figure supplement 1D–F*). Secondary structures in a transcript can be estimated by prediction of minimum free energy (MFE) (*Lorenz et al., 2011*; *Mathews and Turner, 2006*), which is positively correlated with translation rate and a measure of RNA stability (*Nomura et al., 1984*). Our analysis revealed that the MFE for the CDS as well as the 5' and 3'UTRs were overall higher for module 3, indicative of less stable secondary structures (*Figure 3—figure supplement 1G–I*). In contrast, the 5'UTRs of modules 8 and 9 genes showed overall significantly lower MFE indicating more stable structures (*Figure 3—figure supplement 1H*). Previous studies have shown that neurons are particularly enriched for longer genes that are susceptible to differential transcriptional regulation (*Gabel et al., 2015*; *King et al., 2013*; *McCoy et al., 2018*; *Miura et al., 2013*; *Sugino et al., 2014*). Our analysis supports these observations and additionally raises the possibility that the long transcripts of neuron-enriched genes and their complex UTRs might also contribute to complex posttranscriptional regulatory mechanisms. On the contrary, short and simpler transcripts in the ribosome-enriched module 3 might circumvent some of the gene regulatory mechanisms and govern their unique expression pattern (*Figure 3—figure supplement 1K*).

Translation initiation efficiency is directly influenced by the Kozak sequence, which refers to the six nucleotides preceding the AUG start codon and the nucleotide immediately downstream. We computationally calculated a Kozak context score (*Floor and Doudna, 2016*) that describes the overall similarity to the consensus sequence GCCRCCAUGG (*R*=A or G) for all transcripts. Interestingly, genes in 'C' modules have weaker Kozak context scores, suggesting reduced translational initiation rates that potentially allow precise translational regulation of these transcripts via additional mechanisms (*Figure 3D*). In contrast, genes in 'A' modules have the overall highest Kozak scores (*Figure 3D*). Similar differences between 'C' and 'A' modules were also observed for the occurrence of upstream open reading frames (uORFs) in the 5' UTR. It has been suggested that uORFs do not affect mRNA levels but can significantly reduce translation (*Barbosa et al., 2013*). We predicted uORFs by occurrence of [ACT]TG sequence upstream of the main ORF (*Floor and Doudna, 2016*). We observed a reduced number of uORFs predicted in modules 3, 4, and 5 (*Figure 3E*), suggesting a limited role of uORFs in their expression pattern in contrast to gene expression in the 'C' modules. Lastly, we observed that the 3'UTRs of genes in module 6 are enriched in adenylate/uridylate (AU)-rich sequences that promote cellular context-dependent mRNA decay or stability (*Figure 3—figure supplement 1J*; *Otsuka et al., 2019*). This mode of regulation probably supports the expression pattern of this module, where transcripts are translationally silenced in late progenitors and active in neurons (*Figure 2B*). On the other hand, module 9 has significantly less AU-rich elements in the 3'UTR (*Figure 3—figure supplement 1J*) that corroborates the observed depletion of binding motifs for the ELAV-like family (*Figure 3C*), which

is a major binding partner of AU-rich sequences (*Ma et al., 1997*). Overall, this analysis revealed that cell class-specific gene expression seems to be regulated by diverse posttranscriptional mechanisms: trans-regulatory factors, secondary structures in UTRs, imperfect Kozak and uORFs (*Figure 3—figure supplement 1K*). These mechanisms might contribute to the high correlation of relative RNA and protein abundances. However, despite having perfect Kozak sequences, 'A' module genes have the lowest correlation of relative levels of RNA and protein (*Figure 2—figure supplement 2B*).

GO-term analysis showed that 'A' modules (modules 3 and 5) were enriched in ribosome and ribosomal biogenesis genes (*Figure 2C*, *Figure 2—figure supplement 1A and B*). This was especially striking for module 3 which was enriched in ribosomal proteins and eukaryotic translation initiation factors that contain the unique regulatory 5'TOP motif in their 5'UTR (*Figure 3F and G*; *Figure 3— source data 3*; *Cockman et al., 2020*; *Philippe et al., 2020*; *Yamashita et al., 2008*). This observation raised the possibility that 5'TOP genes exhibit a unique RNA-protein expression pattern during corticogenesis.

## Relative protein yield of 5'TOP mRNAs is lower in early progenitors compared to early born neurons

Analyzing the expression pattern of all 5'TOP genes showed that although expressed highly in both progenitors and neurons, 5'TOP transcripts were relatively more abundant in progenitors compared to neurons, with this effect being the most striking at early stage of organoid development at day 40 (*Figure 3H*). Analyzing the transcript distribution by RNA-FISH for two example 5'TOP genes *RPL5 and RPL11* in 40-day-old organoids confirmed this observation (*Figure 4A*), with high intensity of fluorescence being present in the progenitor-rich VZ, and low in the surrounding neuron-rich region. This pattern was not observed by RNA-FISH for the immature neuron marker *DCX* and in the no-probe control (*Figure 4—figure supplement 1A and B*). At protein level, however, a relative difference in the immunostaining in progenitor- and neuron-rich regions was not detectable (*Figure 4B*). This discrepancy was also replicated in the absolute levels of RNA and protein for RPL5 and RPL11 (data available in the R shiny app), indicating that it is not an artifact of the normalization strategy. To verify this RNA-protein discrepancy at cellular resolution, we grew sparsely labeled organoids using 95% of unlabeled control H9 cells and 5% of the dual reporter cell line (*Figure 1—figure supplement 2A*). Next, we stained cryosections from these organoids for transcripts and protein of the 5'TOP genes *RPL5* and *RPL11* by RNA-FISH and by immunostainings, respectively. We quantified the average RNA and protein intensity per cell of individual progenitors labeled by EGFP and neurons labeled by dTomato (*Figure 4—figure supplement 1C*). Indeed, we could validate that *RPL5* and *RPL11* show higher transcript levels in progenitors than neurons, but protein levels were not different between the two cell classes (*Figure 4—figure supplement 1D and E*). To confirm that this observation is not brain organoid-specific and is also found in vivo, we analyzed a published human fetal brain scRNA-seq dataset (*Eze et al., 2021*). Similar to our results, 5'TOP transcripts show higher expression in neuroepithelial, radial glia and intermediate progenitor cells compared to neurons at gestational weeks 6–10 (*Figure 4—figure supplement 1F*).

To investigate possible molecular mechanisms that contribute to the observed discrepancy between RNA and protein levels of the 5'TOP genes, we analyzed a published ribosome profiling dataset from in vitro 2D cultures of hPSC-derived human neural progenitors and neurons (*Blair et al., 2017*; *Figure 4—figure supplement 1G*). This dataset based on WIBR3 hESCs corroborated our observation that progenitors have higher levels of cytosolic 5'TOP mRNAs compared to neurons. Interestingly, while for 5'TOP mRNAs the actively translated (polysome-bound) mRNA pool was similar between the cell classes, progenitors contained significantly higher amounts of monosome-bound mRNA that is considered translationally less active (*Figure 4—figure supplement 1H*). In contrast, transcripts of cell class-specific modules 1 and 9 showed not only a cell class-specific expression in the cytosol but also coherently their specific occurrence in the translated polysome fraction (*Figure 4—figure supplement 1H*). Hence, we hypothesized that some of the excess 5'TOP transcripts present in the progenitors might reside in a translationally inhibited stage.

To test if similar mechanisms operate in the 3D organoid tissue and if the translation of 5'TOP genes is indeed different between early progenitors and neurons, we generated hPSC lines (H9 and Rozh-5 backgrounds) carrying a reporter that contained a 5'TOP motif in the 5'UTR of a doxycycline-inducible tagBFP fluorescent protein. As a negative control we generated non-5'TOP control reporter

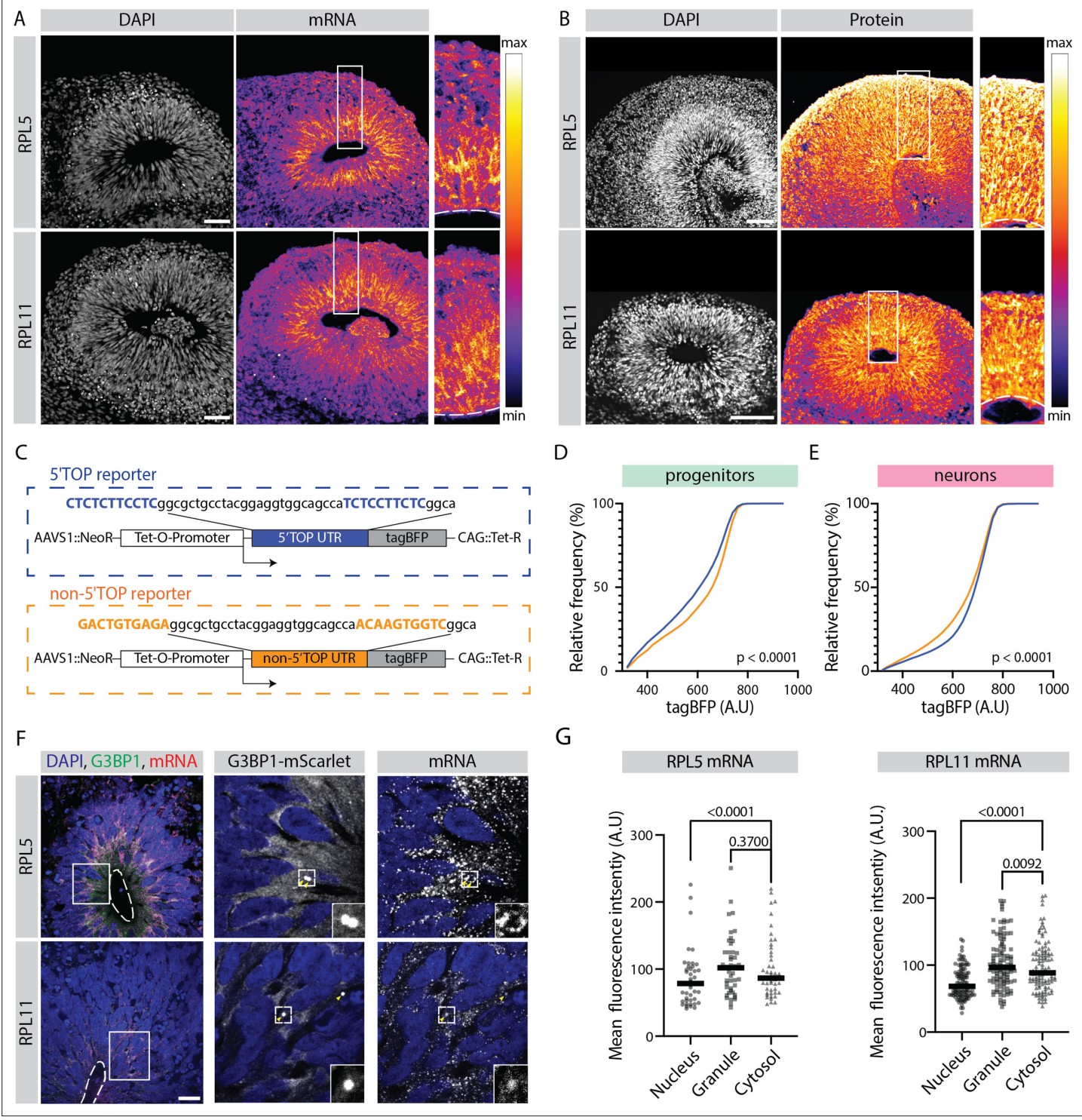

**Figure 4.** Translation of 5' terminal oligopyrimidine (5'TOP) mRNAs is partially inhibited in early progenitors compared to early born neurons. (**A**) RNA-FISH for the 5'TOP transcripts RPL5 and RPL11 in 40-day-old organoids (H9 background). Images show a typical ventricular zone (VZ)-like structure in the brain organoid with a zoomed-in image of the boxed area on the right. Dotted line marks the apical side of VZ. Scale bar = 50 µm. (**B**) Immunostaining for the 5'TOP proteins RPL5 and RPL11 in 40-day-old organoids (H9 background). Images show a typical VZ-like structure in the brain organoid with a zoomed-in image of the boxed area on the right. Dotted line marks the apical side of VZ. Scale bar = 50 µm. (**C**) Schematic representation of the dox-inducible 5'TOP (blue) and non-5'TOP (orange) tagBFP reporter constructs. (**D,E**) Representative cumulative distribution of reporter tagBFP levels in neural progenitors (**D**) and in neurons at day 40 (**E**) from 5'TOP reporter (blue) and non-5'TOP reporter (orange). (**F**) Confocal scan of a typical VZ-like structure in a 40-day-old organoid (H9 background) expressing G3BP1-mScarlet stained for RPL5 and RPL11 transcripts by RNA-FISH. Zoomed-in image

*Figure 4 continued on next page*

*Figure 4 continued*

of the inlay shows G3BP1 granules pointed by yellow arrows. Dotted line marks the apical side of VZ. Scale bar = 20 µm. (**G**) Quantification of RPL5 and RPL11 RNA-FISH signal intensities in the nucleus, G3BP1-positive granules and cytosol of progenitors in the VZ. p-Value of Mann-Whitney test. Each dot represents a cell. (Number of cells analyzed: RPL5: n=45; RPL11: n=34.)

The online version of this article includes the following source data and figure supplement(s) for figure 4:

**Figure supplement 1.** Relative protein yield of 5' terminal oligopyrimidine (5'TOP) mRNAs is lower in early progenitors compared to early born neurons.

**Figure supplement 2.** Doxycycline (Dox)-inducible 5' terminal oligopyrimidine (5'TOP)-reporter hPSC lines.

**Figure supplement 3.** 5' Terminal oligopyrimidine (5'TOP) reporter assay in 40-day-old brain organoids.

**Figure supplement 4.** Early ventricular radial glia exhibit stress granule-like structures.

**Figure supplement 5.** Stress granule-like structures in early ventricular radial glia of G3BP1-mScarlet report line organoids.

**Figure supplement 5—source data 1.** Genotyping PCR of G3BP1-mScarlet cell line.

**Figure supplement 6.** Analysis of relative mRNA stability.

**Figure supplement 6—source data 1.** Relative RNA stability.

lines, in which the 5'TOP nucleotides were mutated (***Figure 4C***). Initially, the fidelity of these reporter lines was evaluated at hPSC stage (***Figure 4—figure supplement 2A***). Untreated hPSCs did not show leaky tagBFP expression (***Figure 4—figure supplement 2B***), whereas doxycycline treatment induced strong tagBFP fluorescence in all six reporter cell lines and no differences between the 5'TOP and non-5'TOP reporters were observed (***Figure 4—figure supplement 2B***). As the translation of 5'TOP mRNAs is known to be inhibited upon mTOR inhibition (***Cockman et al., 2020***), we treated cells with the mTOR inhibitor everolimus. As expected, we observed reduced tagBFP expression in the 5'TOP compared to the non-5'TOP reporter lines (***Figure 4—figure supplement 2C***). This observation confirmed that tagBFP expression in the reporter lines is indeed mTOR pathway-regulated and thus suitable to study 5'TOP translation during organoid development. Hence, we used these reporter lines to grow organoids that we treated at day 40 with doxycycline to induce the expression of tagBFP. Three days after doxycycline induction, the organoids were dissociated to make a single cell suspension and immunostained for progenitor (SOX2) and neuron (TUJ1) markers (***Figure 4—figure supplement 3A***). The expression of tagBFP and cell class-specific markers was analyzed by flow cytometry (***Figure 4—figure supplement 3B–C***). We observed that only ~30% of the cells in the organoids expressed tagBFP, indicating that it is difficult to achieve uniform doxycycline-mediated activation of the construct in 3D organoid tissue (***Figure 4—figure supplement 3B***). We therefore only analyzed cells that expressed tagBFP and were SOX2-PE or TUJ1-488 positive (***Figure 4—figure supplement 3C***). In progenitors (SOX2+), the 5'TOP reporter showed more cells with lower tagBFP fluorescence compared to the non-5'TOP reporter (***Figure 4D***). On the other hand, in neurons (TUJ1+), the 5'TOP reporter showed the tendency toward higher tagBFP expressing cells compared to the non-5'TOP reporter (***Figure 4E***). This was observed consistently between different human pluripotent stem cell lines (H9 and Rozh-5 backgrounds) and experiments (***Figure 4—figure supplement 3D and E***), indicating that in general, the presence of 5'TOP motif lowers the effective tagBFP protein yield in progenitors. Together, we provide multiple pieces of evidence indicating that despite higher mRNA levels, 5'TOP element-containing mRNAs do not give higher protein yield in early progenitors than neurons.

## Early progenitors exhibit developmental use of a stress-associated 5'TOP translational control mechanism

5'TOP translation is mainly known to be regulated during cellular stress (***Cockman et al., 2020***). During stress conditions, 5'TOP RNAs localize to stress granules (SGs), which are dense RNA-protein condensates (***Damgaard and Lykke-Andersen, 2011***; ***Wilbertz et al., 2019***). Hence, we hypothesized that a similar mechanism might play a role during corticogenesis in organoids and asked if we observe stress granule-like (SGL) structures in the brain organoid tissue. We immunostained for the SG marker G3BP1 in untreated organoids and organoids treated with sodium arsenite (NaAs), a strong inducer of cellular stress. As expected, more G3BP1-positive puncta (***Figure 4—figure supplement 4A***) with higher fluorescence intensity (***Figure 4—figure supplement 4B***) were observed in

the NaAs-treated organoids. However, the occurrence of weak G3BP1 puncta in control organoids, particularly in the VZ was supportive of our hypothesis (*Figure 4—figure supplement 4C*). Similar G3BP1-positive puncta were also observed in the VZ of organoids grown from the Rozh-5 hPSC line (*Figure 4—figure supplement 4D*). Analyzing the distribution of another stress marker phospho-EIF2 confirmed that unlike NaAs-treated organoids, the control organoids did not show generally high levels of phospho-EIF2. This observation showed that the control organoid tissue is not under acute stress (*Figure 4—figure supplement 4E and F*).

Next, we asked if 5'TOP transcripts localize to G3BP1-positive SGL structures. For this, we generated a G3BP1 reporter line (H9 background) in which we endogenously tagged G3BP1 with the fluorescent protein mScarlet (*Figure 4—figure supplement 5A*). We generated organoids from this line and observed mScarlet-positive puncta that co-stained with an anti-G3BP1 antibody (*Figure 4—figure supplement 5B*) mostly in the VZ (*Figure 4—figure supplement 5C*). Furthermore, we performed RNA-FISH and observed localization of *RPL5* and *RPL11* transcripts to these mScarlet-positive SGL structures in the VZ (*Figure 4F*). The intensity of the FISH signal was similar in the SGL structures and the cytoplasm (*Figure 4G*), which would suggest that only a small fraction of the mRNA is sequestered in the granules, in agreement with reduced protein yield and not a translational block. In contrast, RPL5 and RPL11 transcripts were strongly enriched in G3BP1-positive granules in NaAs-treated organoids (*Figure 4—figure supplement 5D*) in line with previous studies (*Damgaard and Lykke-Andersen, 2011*; *Wilbertz et al., 2019*).

To reinforce our observation that 5'TOP transcripts are partially translationally silenced, we estimated the relative RNA stability from the fold change of exonic to intronic reads using our RNA-seq data (*Alkallas et al., 2017*; *Figure 4—figure supplement 6—source data 1*). This analysis revealed that 5'TOP RNAs are more stable in the progenitors than neurons (*Figure 4—figure supplement 6A*) and that this difference is most prominent at day 40 (*Figure 4—figure supplement 6B*). In contrast, the cell class-specific modules 1 and 9 showed higher transcript stabilities in progenitors and neurons, respectively, in line with our transcript and protein expression data (*Figure 4—figure supplement 6B*). Thus, early progenitors seem to have excess mature 5'TOP mRNAs with higher RNA stability and a fraction of these is not actively translated and sequestered into G3BP1-positive SGL structures.

The mTOR pathway is a major regulator of 5'TOP translation, which is inhibited upon mTOR pathway inhibition (*Cockman et al., 2020*). Therefore, we assessed the status of the mTOR pathway by immunostaining for phosphorylated forms of two of its substrates: 4E-BP1 and RPS6 (S6). The uniform distribution of phospho-4E-BP1 (p4E-BP1) across the tissue indicated generally active mTOR signaling (*Figure 5—figure supplement 1A*). However, as reported previously for organoid and fetal samples (*Andrews et al., 2020*; *Eze et al., 2021*) phospho-S6 (pS6) staining showed enrichment outside the VZ in neuronal regions (*Figure 5—figure supplement 1B*). These observations suggest that differential mTOR activity in early progenitors, especially ventricular radial glia, and neurons might regulate differential 5'TOP translation. Based on these results we propose that the mTOR-5'TOP translational regulation axis is active in early neural progenitors and might play a crucial role in neuronal differentiation.

## mTOR-mediated regulation of 5'TOP mRNA translation is crucial for the fidelity of cortical development

We hypothesized that mTOR pathway overactivation should result in increased translation of ribosomal and other 5'TOP-containing genes in neural progenitors. To upregulate the mTOR pathway during organoid development, we generated hPSCs (H9 and Rozh-5 background) with loss of function mutations in TSC2 (TSC2$^{-/-}$), a negative regulator of mTOR signaling (*Crino, 2016*; *Figure 5—figure supplement 1C–E*). Immunostainings of organoid tissue confirmed the absence of TSC2 protein in TSC2$^{-/-}$ organoids (*Figure 5A* and *Figure 5—figure supplement 1F*). Furthermore, TSC2$^{-/-}$ organoids showed significantly higher signal of pS6 throughout the VZ when compared to the control tissue (*Figure 5B* and *Figure 5—figure supplement 1G*), indicating overactivation of the TSC2-mTOR pathway. To check if increased TSC2-mTOR signaling indeed leads to excess translation of 5'TOP genes, we performed polysome profiling from control and TSC2$^{-/-}$ organoids (Rozh-5 background, *Figure 5—figure supplement 2A*) and sequenced the ribosome-bound transcripts in the monosome (80S), low- and high-polysome fractions (*Figure 5C*). Due to the large amount of material needed for polysome profiling, we used 37-day-old intact organoids that contain a vast majority of cells in

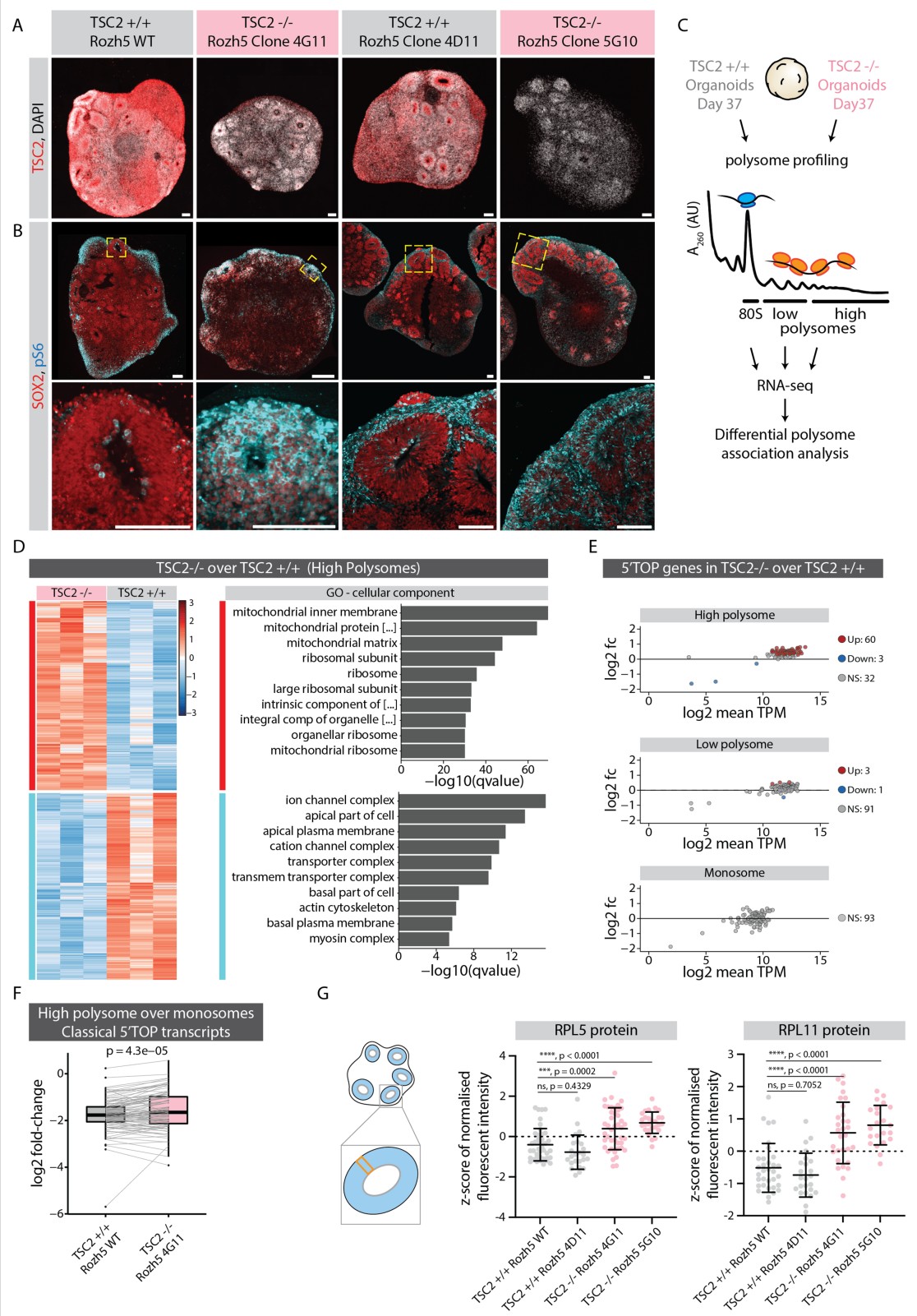

**Figure 5.** mTOR overactivation causes increased translation of 5' terminal oligopyrimidine (5'TOP) transcripts leading to precocious differentiation.
(**A**) Immunostaining of 40-day-old TSC2+/+ and TSC2-/- organoid sections (Rozh-5 background) with anti-TSC2 antibody. Scale bar = 100 μm.
(**B**) Immunostaining of 40-day-old TSC2+/+ and TSC2-/- organoid sections (Rozh-5 background) stained with anti-pS6 and anti-SOX2 antibodies. Zoomed-in image of the inlay on the bottom. Scale bar = 100 μm. (**C**) Schematic representation of polysome profiling of 37-day-old TSC2+/+ and TSC2-/- organoids

*Figure 5 continued*

(Rozh-5 background) followed by RNA-seq and differential polysome association analysis of 80S monosome, low-polysome and high-polysome fractions. Data was generated from three batches of organoid differentiation for Rozh-5 WT and Rozh-5 TSC2$^{-/-}$ 4G11 clones. (**D**) Heatmap of hierarchical clustering of genes differentially associated with high-polysome fractions of TSC2$^{+/+}$ and TSC2$^{-/-}$ organoids (left). Top 10 most significant cellular component gene ontology (GO) terms for TSC2$^{-/-}$ enriched differentially associated genes (DAGs) and TSC2$^{+/+}$ enriched DAGs (right). (**E**) MA plot of 5'TOP genes for high-polysome, low-polysome and monosome fractions shows log2 fold change of TSC2$^{-/-}$ over TSC2$^{+/+}$ versus log2 mean transcript per million (TPM) (expression levels). Significant genes (threshold: log2 fold change = 1, FDR = 0.1) are color coded. (**F**) Log2 fold change of classical 5'TOP genes associated with high-polysome fractions over monosome fractions of TSC2$^{+/+}$ and TSC2$^{-/-}$ organoids. Boxplots mark median and interquartile range (IQR. Whiskers extend to 1.5x IQR). p-Value of paired t-test. (**G**) Quantification of mean z-score normalized fluorescent intensity of RPL5 and RPL11 proteins in the ventricular zones (VZs) of TSC2$^{+/+}$ and TSC2$^{-/-}$ organoids (Rozh-5 background). Error bars mark standard deviation. p-Values of unpaired t-test are shown. Each dot represents a VZ. Data from three batches of organoid differentiation for each clone. Schematic on the left shows a typical regions of interest (ROI) analyzed for a VZ.

The online version of this article includes the following source data and figure supplement(s) for figure 5:

**Source data 1.** Differential gene association analysis from polysome profiling results.

**Figure supplement 1.** Analysis and perturbation of mTOR activity.

**Figure supplement 2.** Polysome profiling in mTOR overactivated organoids.

**Figure supplement 3.** Effect of mTOR overactivation on 5' terminal oligopyrimidine (5'TOP) RNA and protein levels.

**Figure supplement 4.** Effect of mTOR overactivation on tissue development.

the progenitor state. Initially, we performed differential gene association analysis between TSC2$^{-/-}$ and control samples for each fraction (*Figure 5D* and *Figure 5—figure supplement 2B and C*, *Figure 5—source data 1*). We indeed found an enrichment for ribosomal genes specifically in the high-polysome fraction of TSC2$^{-/-}$ over control organoids (*Figure 5D*). When specifically analyzing 5'TOP transcripts we found overall increased levels in the high-polysome fractions in TSC2$^{-/-}$ organoids compared to control organoids, whereas similar levels were detected in the low-polysome and monosome fractions (*Figure 5E*). Accordingly, the relative fraction of 5'TOP transcripts actively translated in high-polysomes over the translationally less active monosomes was increased in TSC2$^{-/-}$ over control organoids (*Figure 5F*, *Figure 5—source data 1*). To confirm that the upregulation of 5'TOP transcripts in the polysome fraction results in increased protein levels, we performed RNA-FISH and immunostainings in TSC2$^{-/-}$ and control organoids for RPL5 and RPL11. While we could not detect differences in their RNA levels (*Figure 5—figure supplement 3A and B*), we observed an increase in protein levels in the VZ of TSC2$^{-/-}$ organoids in H9 as well as Rozh-5 background (*Figure 5G* and *Figure 5—figure supplement 3C and D*). Taken together, these results verify an increase in 5'TOP mRNA translation upon mTOR overactivation and indicate that physiological mTOR activity is crucial to regulate protein levels of 5'TOP genes in early progenitors. Thus, our results suggest that the translational machinery itself is uniquely regulated posttranscriptionally during neurodevelopment.

Next, we asked if mTOR overactivation and the concomitantly increased translation of ribosomes impacts gene regulation of other genes and potentially the tissue development. We checked if the genes differentially associated with ribosome fractions in polysome profiling were enriched in any of the nine gene expression modules (*Figure 2B*) by over- and underrepresentation analysis (*Figure 5—figure supplement 4A and B*). For genes, which were significantly higher in the high-polysome fraction of TSC2$^{-/-}$ over those of control, we observed a significant enrichment for modules 4, 7, and 9 (*Figure 5—figure supplement 4A*). Module 7 is enriched in mitochondrial genes known to be differentially expressed in TSC2$^{-/-}$ at transcript level (*He et al., 2020*; *Koyanagi et al., 2011*). Importantly, the enrichment of neuronal module 9 to be highly translated in TSC2$^{-/-}$ is striking. In contrast, the genes which were less translated in the TSC2$^{-/-}$ organoids over control were significantly enriched in progenitor-specific modules 1 and 2 (*Figure 5—figure supplement 4A*). This data suggests that mTOR overactivation and the subsequent impact on the translation machinery shifts the state of the tissue toward precocious differentiation. These observations were also corroborated in the underrepresentation analysis. Genes with higher translation in TSC2$^{-/-}$ were significantly depleted in modules 1 and 2 (progenitor-specific expression), whereas genes with lower translation in TSC2$^{-/-}$ were significantly depleted in module 9 (neuron-specific expression) as well as module 3, which is enriched in 5'TOP genes (*Figure 5—figure supplement 4B*).

We next tested if the increased translation of differentiation-related genes is reflected by an increased neuronal differentiation using the H9 dual reporter line. For this, we performed flow

cytometry using dual reporter TSC2$^{-/-}$ and TSC2$^{+/+}$ organoids (*Figure 5—figure supplement 4C and D*). In agreement with the polysome profiling data from Rozh-5 organoids, we observed a significant reduction in the fraction of EGFP+ cells (progenitors) and a significant increase in the fraction of dTomato+ cells (neurons) (*Figure 5—figure supplement 4D*). Overall, the ratio of dTomato+ cells to EGFP+ was increased in the TSC2$^{-/-}$, indicative of an increased differentiation (*Figure 5—figure supplement 4D*). Apart from an increase in neuronal markers, we also observed precocious occurrence of the glial marker S100B in the TSC2$^{-/-}$ tissue at day 40 (*Figure 5—figure supplement 4E*), reinforcing our observation that the progenitors are shifted toward a precocious differentiation state. This was already observed at day 40 and possibly explains the premature gliogenesis observed at later stages of TSC2$^{-/-}$ cortical spheroids (*Blair et al., 2018*). While it is difficult to separate the primary and secondary effects of mTOR overactivation, we propose that the partial translational inhibition of 5′TOP mRNAs in early neural progenitors is crucial for cortical development.

## Discussion

In this study, we show that multiple transcripts encoding components of the translational machinery that contain 5′TOP motifs are regulated on a posttranscriptional level during neurodevelopment. 5′TOP transcripts are abundantly expressed in all cells and yet show a unique regulation compared to other cellular transcripts (*Cockman et al., 2020*; *Fonseca et al., 2014*; *Meyuhas and Kahan, 2015*). During mitosis, 5′TOP transcripts skip the global translational inhibition (*Park et al., 2016*), whereas in stress context these transcripts are the first ones to be translationally inhibited to conserve the resources and energy of the cell by downregulating translation (*Cockman et al., 2020*; *Fonseca et al., 2014*; *Meyuhas and Kahan, 2015*). While translational regulation of 5′TOP transcripts is intensely studied in the context of stress, we propose that it also plays a major role during corticogenesis. Our observation of SG-like RNA granules in early ventricular radial glia in human brain organoids suggests possible occurrence of stress-associated processes during human corticogenesis. A recent report suggests that the energetic stage of a cell can impact the formation of SGLs under physiological conditions (*Wang et al., 2022*). Thus, the well-documented difference in the metabolic states of glycolysis-dependent early progenitors and oxidative phosphorylation-heavy neurons (*Iwata and Vanderhaeghen, 2021*) might contribute to the regulation of 5′TOP mRNAs through the formation of SGL structures. Another stress-related pathway, the unfolded protein response (UPR) is physiologically active at early stages of mouse corticogenesis to promote neurogenesis (*Laguesse et al., 2015*). Overactivation of the UPR beyond its physiological levels affects the generation of intermediate progenitors in the mouse and human cortical tissue, resulting in a microcephalic phenotype (*Esk et al., 2020*; *Laguesse et al., 2015*; *Paşca et al., 2019*). Perhaps, similar mechanisms regulating 5′TOP mRNA translation during stress conditions are utilized during development. Recently, a role of the translational regulation of 5′TOP transcripts has been suggested in germline stem cell differentiation in *Drosophila* (*Martin et al., 2022*) and in mouse adult neurogenesis (*Baser et al., 2019*). It will be exciting to investigate if stress-associated pathways play a physiological role during the development of not only the cerebral cortex but also other tissues.

Regulation of the 5′TOP mRNA translation ultimately determines the availability of ribosomes in the cells. Ribosome availability is proposed to be a critical factor determining the translation of mRNAs with complex 5′UTRs with secondary structures (*Hetman and Slomnicki, 2019*). Supporting this hypothesis, we found that module 9 neuronal genes that feature more complex 5′UTRs (*Figure 3—figure supplement 1H*) are particularly deregulated upon 5′TOP translation deregulation accompanying mTOR overactivation (*Figure 5—figure supplement 4A*). Our data suggests that regulating the number of ribosomes is particularly important in the early progenitors. An imbalance of ribosomal components caused by deregulation of 5′TOP transcripts likely causes aberrant translation of differentiation markers and affects the fidelity of cortical development. Reduced ribosome biogenesis and availability is also indicated at an early stage of mouse corticogenesis (*Chau et al., 2018*; *Harnett et al., 2021*). While many gene-specific examples of neural priming in radial glia are known (*Yang et al., 2014*; *Zahr et al., 2018*), our data suggest that radial glia regulate the translation of the translational machinery itself. We speculate that this regulation is crucial to maintain the multipotency of early progenitors to prevent aberrant translation of pre-existing transcripts of differentiation genes. Two new studies have alluded to the role of 5′TOP transcript regulation in the pathophysiology of human 7q11.23 copy variation disorders. These studies also show that perturbed 5′TOP translational

regulation seems to impact the timing of neurogenesis (*Lopez-Tobon et al., 2022*; *Mihailovich et al., 2022*). Thus, more studies show the importance of posttranscriptional regulation of ribosomal proteins in human brain development.

A global regulation of the translational machinery has been recently reported for mid-gestation stages of mouse corticogenesis, with gradual reduction in the translational efficiency of 5'TOP genes and the availability of ribosomes (*Harnett et al., 2021*). It would be interesting to test if similar changes occur during late corticogenesis in humans, which includes a complex repertoire of neural progenitors, especially the outer radial glia.

The mTOR pathway is a major regulator of 5'TOP mRNA translation (*Cockman et al., 2020*; *Fonseca et al., 2014*; *Meyuhas and Kahan, 2015*) and is a key feature of any healthy cell. The role of the mTOR pathway in progenitor proliferation and neuronal morphology and function is well documented (*Switon et al., 2017*). However, the cell type- and developmental stage-specific roles of the mTOR pathway are becoming clear only recently. Our results in human brain organoid tissue show that early radial glia show lower mTOR activity in interphase. Studies using mouse, 2D neuronal cultures and cortical spheroid cultures have also reported a drop in mTOR activity during early stages of corticogenesis (*Blair et al., 2018*; *Blair et al., 2017*; *Chau et al., 2018*). Analysis of human fetal transcriptome of early stages has also suggested a similar trend (*Eze et al., 2021*). On the other hand, human outer radial glia that arise at later stages of corticogenesis exhibit higher mTOR activity, both in primary fetal tissue and in brain organoids (*Andrews et al., 2020*). This indicates that the ventricular and outer radial glia feature different mTOR regulation. In line with this observation, the 5'TOP RNA-protein discrepancy in the progenitors and neurons is reduced at late stages, when the progenitor pool consists of a large population of outer radial glia (*Figure 3H*). Thus, despite its general role, mTOR pathway activity undergoes modulation during distinct neurodevelopmental stages. Additionally, downstream effects of mTOR signaling are linked to diverse cellular processes such as translation, cell cycle, and actin biology. Therefore, linking the defects in the pathway to a final cellular phenotype remains challenging. We provide an alternative molecular role of this pathway to directly regulate ribosome availability and thus indirectly control many other downstream cellular pathways. Mutations in mTOR pathway-related genes are associated with many genetic disorders such as tuberous sclerosis, focal cortical dysplasia, and megalencephaly (*Crino, 2016*). Thus, to understand the disease mechanisms, it will be crucial to consider cell type- and stage-specific regulation of the mTOR pathway.

Beyond 5'TOP genes of the translational machinery, our dataset highlights distinct gene expression modules based on transcript and protein abundance traits. This includes cell class-specific 'C' modules with coherent RNA-protein expression as well as 'T' and 'A' modules where transcript and protein do not follow the same trajectories. Additionally, analysis of cis and trans regulatory factors alludes to distinct RNA regulation mechanisms that are potentially predominant for these genes and contribute to their expression patterns. For instance, the reciprocal relation in the RBP motif enrichment in 3'UTRs of 'C' modules 1 and 9 highlights interesting roles of RBPs to regulate cell class-specific gene expression. Thus, our approach opens new avenues to study RNA regulation during corticogenesis.

Finally, our integrated dataset describes cell class- and developmental stage-specific gene expression, both at RNA and protein level and is available to browse for the wider community in the form of a R-based Shiny app (https://organoid.multiomics.vbc.ac.at). To our knowledge, this is the first time RNA and protein datasets for human corticogenesis have been integrated. Only looking at transcript or protein data wouldn't have revealed the posttranscriptional regulation of the ribosomal genes, which are usually considered housekeeping genes and ignored for example in sc-RNA-seq analysis and often used as normalizers for gene expression. This rich resource can be used to browse RNA-protein expression patterns of various other genes across the human organoid developmental timeline. For instance, our data indicates that the gene regulation of the upper-layer marker SATB2 observed during mouse corticogenesis (*Harnett et al., 2021*) also holds true in humans. As characterized by the general trend of module 6, where the RNA expression of *SATB2* is high in progenitors and neurons of later stages, the trends at protein level suggest translational inhibition of the *SATB2* transcripts in progenitors. Integration with information on protein abundance is crucial for the comprehensive understanding of gene expression. This approach opens a new avenue to study the regulation and expression pattern of individual genes and gene modules. Thus, integrative omics approaches can reveal new biological mechanisms underlying diverse developmental events.

## Limitations of the study

Our study highlights distinct gene expression modules based on transcript and protein abundance traits. To identify common posttranscriptional mechanisms that contribute to observed transcript and protein abundance, this study focused on transcript features influencing translational regulation. In future, it is also crucial to consider additional regulatory mechanisms such as RNA splicing and protein turnover, which were not analyzed in the current study and will contribute to a holistic understanding of posttranscriptional regulation in human corticogenesis.

Our current analysis was performed using human brain organoids, an experimentally amenable system for human corticogenesis. Use of a dual reporter line helped us to assess cell class-specific transcriptomic and proteomic features. Nevertheless, it is important to note that tissue dissociation impacts the morphology of the cells and can cause a bias in the transcripts and proteins sampled in the omics datasets. Hence, it is very crucial to validate the results using intact tissue as done for the 5'TOP motif containing genes in this study. Additionally, future studies are required to validate some of the findings in ethically derived human fetal tissue. Overall, our approach provides a powerful way to propose new hypotheses for posttranscriptional regulation in human corticogenesis.

# Materials and methods

## Plasmid constructs and cloning

All the oligos used in the manuscript are listed in *Supplementary file 1*. All PCRs were performed using Thermo Scientific Phusion Hot Start II DNA-Polymerase (New England Biolabs, M0535S).

## Plasmids for CRISPR-Cas9 gene editing

For cloning guide sequences against SOX2, G3BP1, and TSC2 loci, corresponding DNA oligos containing the guide sequences were phosphorylated and hybridized to make double stranded DNA (ds-DNA) fragments. For SOX2 and TSC2 guides, these ds-DNA fragments were cloned into pSpCas9(BB)-2A-GFP plasmid (Addgene, 48138, RRID:Addgene_48138), modified to express eCas9 instead of WT-Cas9 and dTomato instead of GFP (*Esk et al., 2020*), using the Bbs1 cloning strategy of *Ran et al., 2013*. For G3BP1 guide, the ds-DNA fragments were cloned into pSpCas9(BB)-2A-Puro (PX459) (Addgene, 62988, RRID:Addgene_62988). Both backbones were gifts from Feng Zhang (*Ran et al., 2013*). See *Supplementary file 1* for the guide and oligo sequences.

## Homology plasmids

### SOX2-P2A-EGFP homology construct

Homology construct for tagging SOX2 locus with P2A-EGFP was assembled by Gibson cloning. A vector backbone containing Diphtheria toxin A cassette, as a negative selection marker, was used. Left and right homology arms of the SOX2 locus were amplified from H9 gDNA with primers in which the native stop codon was removed. The P2A-EGFP fragment was amplified with the mentioned primers. The intermediate plasmid was used for mutagenesis PCR to mutate the guide cutting base from G to A to avoid cutting of the repaired clones.

### AAVS1 hSYN1::dTomato homology construct

Donor plasmid was constructed to insert the following cassette SA-2A-puro-PA-2xCHS4-hSYN1-INTRON-tdTomato-WPRE-SV40-2xCHS4. For this purpose, a previously generated backbone containing 2xCHS4-EF1a-tdTomato-SV40-2xCHS4 was used (*Bagley et al., 2017*). The promoter was exchanged with hSYN1 promoter.

### tagBFP reporter constructs (5'TOP reporter and non-5'TOP reporter)

Donor plasmids for doxycycline-inducible 5'TOP and non-5'TOP reporters were generated by Gibson assembly to insert the following cassettes: (1) Neo-TetO-minCMV-5'TOPUTR-INTRON-tagBFP-bGHpA-CAG-TetR and (2) Neo-TetO-minCMV-non-5'TOPUTR-INTRON-tagBFP-bGHpA-CAG-TetR, respectively. pAAVS1-Neo-TRE-CMV-Cre-rtTA, containing the Tet-response element and CAG:Tet repressor sequence, was a gift from Madeline Lancaster and was used as backbone upon digestion with Kpn1 and Sal1 (Addgene, 165457, RRID:Addgene_165457) (*Benito-Kwiecinski et al., 2021*). The

5'TOP sequence of RPL32 with CMV minimal promoter and chimeric intron was amplified from plasmid pSF4-TetCMV-5'TOP-intron-20xGCN4-Renilla-FKBP-Stop-24xMS2v5-SV40-CTE-polyA (Addgene, 19946, RRID:Addgene_119946), which was a gift from Jeffrey Chao (*Wilbertz et al., 2019*). For the non-5'TOP, the pyrimidine-rich sequences in the 5' terminus of the RPL32 sequence were mutated and ordered as a gblock.

## G3BP1 homology construct

The plasmid HR_G3BP1-V5-APEX2-GFP (Addgene, 105284, RRID:Addgene_105284) was a gift from the Eugene Yeo lab (San Diego, CA, USA) (*Markmiller et al., 2018*) and the plasmid pmScarlet-i_C1 was a gift from Dorus Gadella (Addgene, 85044) (*Bindels et al., 2017*). To generate the plasmid pHDR_G3BP1-V5-APEX2-mScarlet-I, used for CRISPR mediated endogenous tagging of G3BP1 with APEX2-mScarlet, the CDS for CopGFP was exchanged with the sequence of the monomeric red fluorescent protein mScarlet (*Bindels et al., 2017*).

## Cell lines and culture

Feeder-free hESCs line WA09 (H9) and feeder-free human induced pluripotent stem cell (hiPSC) line HPSI0114i-rozh_5 (Rozh-5) were used in this study. H9 was obtained from WiCell and Rozh-5 from HipSci. All cells were cultured on hESCs-qualified Matrigel (Corning, 354277) coated plates with a modified in-house medium based on the E8 culture system (*Chen et al., 2011*). The original E8 recipe was supplemented with 0.5% BSA (Europa Bioproducts, EQBAH62-*0500*), 200 ng/ml in-house produced FGF2, and 1.8 ng/ml TGFβ1 (R&D Systems, RD-240-B-010). Cells were passaged every 3–4 days using 0.5 mM EDTA. For generating TSC2$^{-/-}$ knockouts, cells were cultured using Cellartis DEF-CS 500 culture system (Takara Bio, cat. no. Y30012). Cells were verified to display the right identity using STR profiling and a normal karyotype (*Supplementary file 2*). Cells were also regularly tested for mycoplasma contamination.

## Reporter line generation

All reporter lines were generated by nucleofection of the plasmid DNA with the Amaxa nucleofector 2b device (Lonza, AAB-1001) and Human Stem Cell Nucleofector Kit 1 (Lonza, VPH-5012) following the manufacturer's guidelines. For each nucleofection 10$^6$ single cells were used. For genotyping, DNA was extracted using the QuickExtract DNA Extraction Solution (Cambio QE09050) and a PCR assay was performed to identify correctly edited clones. See *Supplementary file 1* for oligo sequences used for genotyping. Genomic integrity of the final clones was confirmed by SNP arrays (*Supplementary file 2*).

Insertion into the AAVS1 safe harbor locus was performed using TALEN technology as described before (*Bagley et al., 2017*; *Hockemeyer et al., 2011*). For this purpose, the nucleofection mix containing 0.5 µg of each of the TALEN plasmids and 1.5 µg of each of the donor plasmids was used. Nucleofected cells were grown for 4–7 days and then selected with appropriate antibiotics. For puromycin resistance reporter: 1 µg/ml puromycin dihydrochloride (Thermo Fisher Scientific, A1113803) and for neomycin resistance reporter 250 µg/ml geneticin/G418 sulfate (Thermo Fisher Scientific, 10131035). Surviving colonies were picked manually, transferred into 24-well plates and further expanded for genotyping and cryopreservation.

## H9 dual reporter line

The dual reporter line was generated in a stepwise manner. First, FF H9 SOX2::SOX2-P2A-EGFP reporter line was generated. For this purpose, the nucleofection mix included 1 µg guide-Cas9 plasmid and 4 µg homology repair construct. 7 days post nucleofection, surviving cells were FACSed to obtain GFP positive cells which were sorted as single cells and further expanded for genotyping and cryopreservation. Positive clones were selected by performing PCR to verify genomic insertion of P2A-EGFP. Using the FF H9 SOX2::SOX2-P2A-EGFP reporter line clone, the dual reporter line was generated by inserting the cassette in the AAVS1 hSYN1::dTomato cassette in the AAVS1 locus as described above. Dual reporter clone 10.1 and clone 10.2 were used for the transcriptome and proteome datasets.

For generating 5'TOP and non-5'TOP reporter lines, FF H9 hESCs and Rozh-5 iPSCs carrying monoallelic insertion of pAAVS1(LH)-SA-2A-puro-7xTetO-minCMV-iRFP-pA-AAVS1(RH) were used.

The reporter constructs were inserted in the AAVS1 locus as described above. Cells resistant to puromycin and neomycin were selected for tag-BFP reporter assay. This strategy ensured monoallelic insertion of the tag-BFP reporter.

For generating G3BP1 reporter line, FF H9 cells were electroporated with 0.8 µg guide-Cas9 plasmid and 1 µg homology repair construct. The HDR plasmid carried in addition to the APEX2-mScarlet fusion, also a floxed puromycin resistance cassette under the control of the EF1a core promoter. 48 hr post electroporation, cells were selected with 0.5 µg/ml puromycin and correct gene editing was confirmed by genotyping PCRs and Sanger sequencing. Following, a pool of the correct clones was nucleofected as before with 2 µg Cre-recombinase-2A-EGFP expressing plasmid and 1 µg of pmaxGFP to remove the EF1a-puromycin cassette. Successfully electroporated cells were sorted by FACS for GFP and mScarlet double positive cells. Excision of the EF1a-puromycin cassette was confirmed by genotyping PCRs and Sanger sequencing.

For generating TSC2 KO hPSCs, Rozh-5 and H9 dual reporter (Clone 10.1) cells were used. Nucleofection was performed with 3 µg guide-Cas9 plasmid. The surviving cells were FACSed the next day for dTomato fluorescence and sorted as single cells into 96-well plates. The single cell clones were expanded and genotyped to identify TSC2 homozygous mutants by Sanger sequencing.

## Dorsal tissue-enriched telencephalic organoid generation

Dorsal forebrain-enriched telencephalic organoids were generated as previously described with slight modifications (*Esk et al., 2020*). Briefly, hPSCs were grown to 60–80% confluency and dissociated with Accutase (Merck, A6964) to obtain a single cell suspension; 4000–9000 cells were resuspended in 150 µl of E8 supplemented with Rock inhibitor and seeded in each well of the 96-well ultra-low-attachment U-bottom plate to form embryoid bodies (EBs). On day 3 the medium was changed to E8. From day 6 onward the EBs were cultured in neural induction medium (NI) (*Lancaster et al., 2017*). On day 10, EBs were embedded in Matrigel and transferred to 10 cm cell-culture dishes coated with anti-adherence rinsing solution (Stemcell Technologies, P2443) in 15 ml NI. From days 13 to 25 the organoids were grown in a differentiation medium lacking vitamin A (Diff-A). During day 13–15 they were treated with two pulses of 3 µM CHIR99021 (Merck, 361571) to dorsalise the tissue. On day 18 dishes were transferred to an orbital shaker at 57 rpm. From day 25 onward the organoids were grown in differentiation medium with vitamin A (Diff+A). The medium was supplemented with 1% Matrigel from day 40 onward and with BDNF from day 55 onward. See the media composition in *Supplementary file 3*.

For Tet-induction, the medium was supplemented with doxycycline-hyclate (Merck, D9891) dissolved in water at a final concentration of 1.7 µg/ml medium of hPSCs and 3 µg/ml for organoids. For mTOR inhibition, hPSC culture medium was supplemented with everolimus (Abcam, ab142151) dissolved in DMSO at a final concentration of 20 nM. For NaAs treatment, the medium was supplemented with NaAs (Merck, S7400) dissolved in water to achieve a final concentration of 500 µM. To ensure diffusion in the 3D organoid tissue, treatment with NaAs was performed for 1 hr.

## Organoid dissociation

Organoids were washed with DPBS without calcium and magnesium and added to a 9:1 mixture of Accutase (Merck, A6954) and 10× Trypsin (Thermo Fisher Scientific, 15400054) supplemented with 2 units/ml TURBO DNase (Thermo Fisher Scientific, AM2238). The dissociation was performed using the NTDK1 protocol on a gentleMACS Dissociator (Miltenyi Biotec, 130-093-235). The dissociated cells were pelleted, washed with DPBS without calcium and magnesium (Thermo Fisher Scientific, 14190250), and filtered through a 70 µm cell strainer to remove residual tissue chunks. The filtered single cell suspension was then used either for live cell FACS or for immunostaining.

## FACS

For live cell sorting cells were resuspended in FACS buffer (DPBS without calcium and magensium with 2% BSA), filtered through a 35 µm cell strainer and then sorted on a FACS ARIAIII (BD Biosciences) controlled by FACSDiva software. For detecting GFP signal, a 488 nm laser was used with a 530/30 nm filter. For detecting dTomato signal, a 561 nm laser was used with 582/15 nm filter. The sort was done in PBS at low pressure using a 100 µm nozzle.

## Immunostaining of dissociated cells

For immunostaining cells, the dissociated single cells were fixed with 4% paraformaldehyde (PFA) for 30 min on ice. The cells were permeabilized with 0.1% Saponin (Merck, 47036) in PBS either during or after fixation. Fixed cells were then pelleted and washed with a wash solution (0.1% Saponin + 1% BSA in PBS) to remove traces of PFA. Next, the cells were split for appropriate controls and resuspended in a staining solution (fluorophore conjugated primary antibody in 0.1% Saponin + 1% BSA in PBS) for 30 min on ice. The stained cells were then pelleted and washed twice with wash solution and resuspended in the final resuspension buffer (0.5% BSA in PBS). Antibodies used in this study are summarized in *Supplementary file 3*.

## Flow cytometry analysis

For analyzing the expression in live cells or in immunostained single cells, flow cytometry was performed on LSR Fortessa (BD Biosciences) controlled by FACSDiva software. Data was analyzed using FlowJo. For detecting tagBPF signal, 405 nm laser was used with 442/46 nm filter. For detecting Alexa-488 signal, 488 nm laser was used with 530/30 nm filter. For detecting PE signal, 561 nm laser was used with 582/15 nm filter.

## Immunohistochemistry

Organoids were fixed in 4% PFA for 3–4 hr at room temperature. After extensive washes with PBS, organoids were immersed overnight, first in 30% sucrose (Merck, 84097) in PBS and thereafter in 1:1 mixture of 30% sucrose and OCT (Sakura, 4583). These organoids were then embedded in OCT using suitable cryomolds and frozen at –70°C. Samples were sectioned at 20 µm thickness using a Epredia Cryostar NX70 cryostat (Thermo Fisher, 957000H). The slides were dried overnight and then stored at –20°C.

After washing the slides extensively with PBS to remove the OCT, cryo-sections were permeabilized and blocked with blocking solution (5% BSA containing 0.3% Triton X-100 [Merck, 93420]) for 1–2 hr at room temperature. Sections were then incubated with primary antibodies diluted in antibody solution (5% BSA, 0.1% Triton X-100 in PBS) overnight at 4°C. After three washes of 10 min with PBST (0.01% Triton X-100 in PBS), sections were incubated with secondary antibodies diluted in antibody solution containing 2 µg/ml DAPI at room temperature for 2 hr. After three washes of 10 min with PBST (0.01% Triton X-100 in PBS) and one wash with PBS, the samples were mounted in fluorescence mounting medium (Agilent, S302380-2) and imaged with a spinning disk confocal microscope. Antibodies used in this study are summarized in *Supplementary file 3*. AlexaFluor 488-, 568-, or 647-conjugated secondary donkey antibodies were used at a 1:500 dilution.

## RNA-FISH

FISH probes for *RPL5, RPL11* (*Wilbertz et al., 2019*) and *DCX* (Stellaris, VSMF-2504-5) were ordered from Stellaris. RPL5 and RPL11 probes were labeled with Quasar 570 Dye. Sequences of the FISH-probes and corresponding fluorophores are listed in *Supplementary file 1*. DCX probes were labeled with Quasar 670 Dye. Slides with cryosections were thawed and washed twice with DEPC-treated PBS, once with nuclease-free water and then immersed in a 1× TEA buffer (triethylammonium acetate buffer) for 10 min. For permeabilization, sections were immersed in 2× SSC buffer and subsequently in 70%, 95%, and 100% ethanol. After air drying for 1.5 hr, sections were incubated in hybridization solution (10% dextran sulfate [Merck, S4030], 1 mg/ml *Escherichia coli* tRNA [Merck, 10109541001], 2 mM vanadyl ribonucleoside complex [New England Biolabs, S1402S], 2× SCC, 10% formamide [Merck, F9037], 0.5% BSA) containing 250 nM of the FISH probe at 37°C overnight. After incubation, the slides were washed with wash buffer A (Stellaris, SMF-WA1-60) for 30 min at 37°C and subsequently stained with Hoechst for 30 min at 37°C in the dark. Next, the slides were washed with wash buffer B (Stellaris, SMF-WB1-20), subsequently immersed in 50%, 85%, and 100% ethanol for 3 min and air dried. Finally, the samples were mounted in fluorescence mounting medium (Agilent, S302380-2) and imaged with a spinning disk confocal microscope.

## Imaging and image analysis

Fluorescence images of immunostainings and RNA-FISH were obtained on an Olympus spinning disk confocal based on an Olympus IX3 Series (IX83) inverted microscope, equipped with a dual-camera

Yokogawa W1 spinning disk (SD) allowing fast confocal acquisition. All components were controlled by CellSense software. Objectives used with the spinning disk confocal were 4×/0.16 (Air) WD 13 mm, 10×/0.4 (Air) WD 3.1 mm (for organoid overview), 20×/0.8 (Air) WD 0.6 mm (for imaging individual VZs), 40×/1.25 (Silicon Oil) WD 0.3 mm, and 100×/1.45 (Oil) WD 0.13 mm (for imaging SGL).

Preparation of image panels and image analysis was performed using Fiji (*Schindelin et al., 2012*). For measuring average intensity per cell, progenitors and neurons were identified in mosaic reporter organoids, and the cell outline was traced to mark a ROI. Average intensities for each ROI were measured using the multi-measure tool. To analyze the distribution of G3BP1-mScarlet granules, (a) the number of granules in the VZ and neuronal zone was counted per field. The distribution was calculated as granule density, the number of granules was normalized to the number of nuclei in the field and as (b) the percentage of granules within the VZ per field. For measuring G3BP1 and FISH signal intensity in the granule-like structures, ROIs were drawn by tracing the outline of the granules. For measuring FISH signals in the nucleus and cytoplasm, similar ROIs were drawn in the nucleus and cytoplasm. Average intensities for each ROI were measured using the multi-measure tool. For measuring average antibody staining intensity in the VZ, a rectangular ROI was marked in the maximum intensity projection of the VZ. Average intensities for each ROI were measured using the multi measure tool. To enable integration of data from different experiments, results were z-score normalized per experiment.

## RNA extraction, library generation, and RNA-seq

FACSed cells were pelleted and used for RNA isolation. Total RNA was isolated using a lysis step based on guanidine thiocyanate (adapted from *Boom et al., 1990*) with DNaseI digestion using magnetic beads (GE Healthcare, 65152105050450) applied on the KingFisher instrument (Thermo Fisher Scientific). The purified RNA was used to generate the RNA-seq library.

RNA sequencing libraries were generated using 500 ng of RNA using NEBNext Ultra II Directional RNA Library Prep Kit for Illumina (New England Biolabs, E7760). Isolation of mRNA was done using the NEBNext Poly(A) mRNA Magnetic Isolation Module (New England Biolabs, E7490). The size of libraries was assessed using NGS HS analysis kit and a fragment analyzer system (Agilent). Library concentrations were quantified with KAPA Kit (Roche) and 75 bp paired end-read sequencing was performed using the Illumina NextSeq550 platform.

RNA-seq reads were trimmed using trim-galore v0.5.0 and reads mapping to abundant sequences included in the iGenomes UCSC hg38 reference (human rDNA, human mitochondrial chromosome, phiX174 genome, adapter) were removed using bowtie2 v2.3.4.1 alignment. Remaining reads were analyzed using genome and gene annotation for the GRCh38/hg38 assembly obtained from *Homo sapiens* Ensembl release 94. Reads were aligned to the genome using star v2.6.0c and reads in genes were counted with featureCounts (subread v1.6.2). Differential gene expression analysis on raw counts and variance-stabilized transformation of count data for heatmap visualization were performed using DESeq2 v1.18.1.

Published scRNA-seq dataset (*Eze et al., 2021*) was analyzed using Seurat package (*Stuart et al., 2019*) to obtain 5′TOP module scores in fetal cell types.

## Proteomics

A minimum of $2×10^6$ FACSed cells were pelleted, snap frozen in liquid nitrogen, and stored at –80°C until processed. Each cell pellet was resuspended in 72.5 µl 10 M urea 50 mM HCl and incubated 10 min at room temperature. Then, 7.5 µl 1 M TEAB (triethylammonium bicarbonate) buffer (pH 8) was added and total protein amount determined by Bradford assays. Following, 1 µl benzonase and 1.25 µl 1 M DTT (dithiothreitol) was added to each sample and incubated for 37°C for 1 hr. Protein alkylation was performed with 2.5 µl of 1 M IAA (iodoacetamide) and incubated for 30 min in the dark at room temperature and the reaction quenched with 0.6 µl 1 M DTT. All samples were diluted with 100 mM TEAB buffer to a urea concentration of 6 M and proteins digested with LysC (Wako) for 3 hr at 37°C using an enzyme to protein ratio of 1:50. After the first digestion with LysC the samples were diluted with 100 mM TRIS buffer to a final urea concentration of 2 M. The final tryptic digestion (Trypsin Gold, Promega) was performed with an enzyme to protein ratio of 1:50 overnight at 37°C. To evaluate the digest efficiency 100 ng of all samples were analyzed on a monolithic HPLC system. Following, peptides were desalted using C18 cartridges (Sep-Pak Vac 1 cc [50 mg], Waters) that were

equilibrated with 500 µl methanol/water 50:50, 2×0.5 ml 80% acetonitrile (ACN) 0.1% formic acid (FA), and 3×0.5 ml 0.1% trifluoroacetic acid (TFA) applying gentle pressure with compressed air. The pH of all samples was adjusted to pH <2 with 10% TFA before they were loaded on the columns. All columns were washed with 6×0.5 ml 0.1% TFA applying gentle pressure using compressed air. To elute the sample 2×200 µl of 80% ACN and 0.1% FA were used and to remove the solvent completely from the column, gentle pressure was applied with compressed air. The organic content of the eluates was removed by using a SpeedVac vacuum concentrator and samples were diluted to a volume of 200 µl with 0.1% FA, snap frozen with liquid nitrogen, and lyophilized overnight. The dried samples were dissolved in 100 µl 100 mM HEPES and quantified using a monolithic HPLC system. One bridge channel was produced from all FACS sorted cells. In these channels the corresponding samples were mixed in a 1-1 ratio.

For TMT labeling, 20 µg of peptides (in 100 µl 100 mM HEPES pH 7.6) were labeled with distinct channels of the TMTpro16 plex reagent (Thermo Fisher, TMT16plex) according to the manufacturer's description. Labeling efficiency was determined by LC-MS/MS on a small aliquot of each sample. Samples were mixed in equimolar amounts and equimolarity was evaluated again by LC-MS/MS. The mixed sample was acidified to a pH below 2 with 10% TFA and was desalted using C18 cartridges (Sep-Pak Vac 1 cc [50 mg], Waters). Peptides were eluted with 3×150 ml 80% acetonitrile (ACN) and 0.1% FA, followed by freeze-drying. The dried samples were dissolved in 70 µl of SCX buffer A (5 mM NaH$_2$PO$_4$, pH 2.7, 15% ACN) and 200 µg of peptide were loaded on the column. SCX was performed using a custom-made TSK gel SP-2PW SCX column (5 µm particles, 12.5 nm pore size, 1 mm ID × 250 mm, TOSOH) on an Ultimate system (Thermo Fisher Scientific) at a flow rate of 35 µl/min. For the separation, a ternary gradient was used starting with 100% buffer A for 10 min, followed by a linear increase to 10% buffer B (5 mM NaH$_2$PO$_4$, pH 2.7, 1 M NaCl, 15% ACN) and 50% buffer C (5 mM Na$_2$HPO$_4$, pH 6, 15% ACN) in 10 min, to 25% buffer B and 50% buffer C in 10 min, 50% buffer B and 50% buffer C in 5 min and an isocratic elution for further 15 min. The flow-through was collected as single fractions, along with the gradient fractions that were collected every minute. In total 60 fractions were collected, and low abundant fractions were pooled. ACN was removed by vacuum centrifugation and the samples were acidified with 0.1% TFA and analyzed by LC-MS/MS. The UltiMate 3000 HPLC RSLC nano system (Thermo Scientific) was coupled to a Q Exactive HF-X mass spectrometer (Thermo Scientific), equipped with a Proxeon nanospray source (Thermo Scientific). Peptides were loaded onto a trap column (Thermo Scientific, PepMap C18, 5 mm×300 µm ID, 5 µm particles, 100 Å pore size) at a flow rate of 25 µl/min using 0.1% TFA as mobile phase. After 10 min, the trap column was switched in line with the analytical column (Thermo Scientific, PepMap C18, 500 mm×75 µm ID, 2 µm, 100 Å). Peptides were eluted using a flow rate of 230 nl/min and a binary 120 min gradient. The two-step gradient started with the mobile phases: 98% A (water/FA, 99.9/0.1, v/v) and 2% B (water/acetonitrile/FA, 19.92/80/0.08, v/v/v) increased to 35% B over the next 120 min, followed by a gradient in 5 min to 90% B, stayed there for 5 min and decreased in 2 min back to the gradient 95% A and 2% B for equilibration at 30°C.

The Q Exactive HF-X mass spectrometer was operated in data-dependent mode, using a full scan (m/z range 375–1650, nominal resolution of 120,000, target value 3e6) followed by MS/MS scans of the 10 most abundant ions. MS/MS spectra were acquired using normalized collision energy of 35, isolation width of 0.7 m/z, resolution of 45,000, a target value of 1e5, and maximum fill time of 250 ms. For the detection of the TMT reporter ions a fixed first mass of 110 m/z was set for the MS/MS scans. Precursor ions selected for fragmentation (exclude charge state 1, 7, 8, >8) were put on a dynamic exclusion list for 60 s. Additionally, the minimum AGC target was set to 1e4 and intensity threshold was calculated to be 4e4. The peptide match feature was set to preferred and the exclude isotopes feature was enabled.

## Proteomics data processing

For peptide identification, the RAW-files were loaded into Proteome Discoverer (version 2.4.0.305, Thermo Scientific). All hereby created MS/MS spectra were searched using MSAmanda v2.0.0.13248 (*Dorfer et al., 2014*). The RAW-files were searched against the human uniprot-reference-database (20,541 sequences; 11,395,640 residues). The following search parameters were used: Iodoacetamide derivative on cysteine was set as fixed modification, deamidation on asparagine and glutamine, oxidation on methionine, TMTpro-16plex TMT on lysine, phosphorylation on serine, threonine and tyrosine

as well as carbamylation on peptide N-terminus, TMTpro-16plex TMT on peptide N-terminus, acetylation on protein N-terminus were set as variable modifications. Monoisotopic masses were searched within unrestricted protein masses for tryptic enzymatic specificity. The peptide mass tolerance was set to ±5 ppm and the fragment mass tolerance to ±15 ppm. The maximal number of missed cleavages was set to 2. The result was filtered to 1% FDR on peptide-spectrum-match and protein level using Percolator (*Käll et al., 2007*) algorithm integrated in Thermo Proteome Discoverer. The localization of the modification sites within the peptides was performed with the tool ptmRS, which is based on phosphoRS (*Taus et al., 2011*). Peptides were quantified based on Reporter Ion intensities extracted by the 'Reporter Ions Quantifier'-node implemented in Proteome Discoverer. Proteins were quantified by summing reporter intensities of spectra with less than 50% isolation interference from unique peptides. Inspired by the iBAQ algorithm (*Schwanhäusser et al., 2011*), protein abundances were normalized by dividing summarized reporter intensities by the number of theoretically observable tryptic peptides with a length between 7 and 30 amino acids. TMT batches were normalized using a 'bridge-channel' as reference which contains a mix of all samples.

We used a previously described graphical formula to identify significant proteins in volcano plots (*Hein et al., 2015*): $-\log10\left(p\right) \geq \frac{c}{|x|-x_0}$ with 'x: enrichment factor of a protein'; 'p: p-value of the protein-wise linear models and empirical Bayes statistics, calculated from replicates using the R bioconductor package DEP'; '$x_0$: fixed minimum enrichment' and 'c: curvature parameter', choosing c=3.65 and $x_0$=1.75 for 1% FDR.

## Integration of transcriptomic and proteomic dataset

For the 6740 UniProt protein identifiers from the proteomics dataset, 6732 could be mapped to an ensembl gene identifier using the BioMart data mining tool as well as manual annotation. Accordingly, for 20,158 out of 33,849 ensembl gene identifiers from the transcriptomic dataset a UniProt protein identifier could be identified. For data integration, the theoretically observable tryptic peptide normalized protein abundances were log10 transformed followed by z-score normalization. TPM values for genes that encode for identical proteins were summed, log10 transformed, and finally z-score normalized. Finally, the normalized proteomics and transcriptomics datasets were combined by UniProt identifiers resulting in gene expression data for 6714 genes.

To identify temporal gene expression modules, the R bioconductor package maSigPro version 1.68.00 for time-series was used (*Nueda et al., 2014*). Briefly, a first regression fit for each of the 6714 genes was performed and significant genes selected at a false discovery rate of 0.05. For the remaining 5978 significant genes, a second stepwise regression fit was performed to identify profile differences between experimental groups. Regression models at R-squared>0.6 were found for 3668 genes, which were hierarchical clustered. The optimal number of clusters was estimated using the factoextra package for R and the 'average silhouette width', 'total within sum of square', and 'gap statistics' methods. Finally, different numbers of clusters (6–12) were manually evaluated with regards to their biologically meaningfulness using for example gene enrichment analysis tools. Finally, the 3668 genes were clustered into nine gene expression modules and gene enrichment analysis performed using the R bioconductor package clusterProfiler (*Yu et al., 2012*).

## Transcript feature analysis

For each cluster, we considered all the transcripts of the member genes detected in our RNA-seq dataset (mean TPM > 1) for the analysis. For transcript feature analysis, the Ensembl release http://sep2019.archive.ensembl.org/index.html in biomart R package was used to derive lengths of transcripts, CDS, and UTRs. RNA structure features were determined using a Python program available through GitHub at https://github.com/stephenfloor/tripseq-analysis, (*Blair et al., 2017*). The structure was computed using RNALfold from the ViennaRNA package (*Lorenz et al., 2011*) in a 75 nt window.

## Analysis of RBP motif enrichment

Analysis of enrichment or depletion of RBP motifs in 5'UTR and 3'UTR sequences was performed by transcript set motif analysis with k-mer method and standard settings of Transite tool (*Krismer et al., 2020*) (https://transite.mit.edu/). Enrichment was calculated per module over the entire gene-set. Adjusted p-value of 0.05 was used as the significance threshold.

## Analysis of published tripseq data

We analyzed the previously published data for quantified expression values for TrIP-seq polysome profiling (*Blair et al., 2017*, Supplementary Table 4). The data was filtered for neural progenitors (progenitors) and 50-day-old neurons (neurons) as well as for 5'TOP genes or genes in module 1 or 9 and all TPM values were log2 transformed.

## Analysis of relative stability of transcripts

The analysis was performed according to a method that removes the bias from expression level differences. In brief, mRNA stability estimates were calculated using REMBRANDTS (https://github.com/csglab/REMBRANDTS; *Alkallas et al., 2017*). Preprocessing was performed following the CRIES workflow (https://github.com/csglab/CRIES, *Alkallas et al., 2017*) applying hisat2 v.2.1.0 in read alignment, samtools v1.10 with parameters -F 1548 -q 30 in alignment-filtering, htseq v0.11.2 with parameter 'intersection-strict' for exonic read summarization, and with parameter 'union' for intronic read counting. REMBRANDTS.sh with linear biasMode, 0.99 stringency cutoff, Rv3.6.2, and DESeq2 v1.26.0 was run to obtain the per sample differential mRNA stability estimates.

## Polysome profiling by sucrose gradient fractionation

Polysome profiling was performed for three batches of organoid differentiation using Rozh-5 WT and Rozh-5 TSC2$^{-/-}$ 4G11 hiPSC clones. For this procedure, all solutions and materials contained 100 µg/ml cycloheximide (CHX) were DPEC-treated if possible, and were pre-chilled at 4°C for at least 30 min before use. Before harvest, organoids were treated with 100 µg/ml CHX at 37°C for 10 min to arrest ribosomes and washed with ice-cold DPBS without calcium and magnesium (Thermo Fisher Scientific, 14190250). Organoids were then dissociated as described above with Accutase/Trypsin mix supplemented with 100 µg/ml CHX. All cells were pelleted at 1000 × $g$ for 10 min at 4°C. The supernatant was removed, cells were washed with 15 ml ice-cold DPBS without calcium and magnesium and pelleted at 1000 × $g$ at 4°C for 10 min. Next the pellet was resuspended in lysis buffer 20 mM TrisHCl, 30 mM MgCl$_2$, 100 mM NaCl, 1% Triton-X, 100 µg/ml CHX, 0.5 mM DTT, 1 mg/ml heparin, 100 µg/ml RNase inhibitor (Promega, NA2615) and Turbo DNase (Thermo Fisher Scientific, AM2238). Samples were triturated 10 times with a 26 G needle, incubated on ice for 10 min, and spun at 21,000 × $g$ at 4°C for 20 min. 400 µl of lysate was loaded onto the sucrose gradient. Sucrose gradients of 10–50% range were prepared in 20 mM TrisHCl, 5 mM MgCl$_2$, 140 mM NaCl with Gradient master unit (BIOCOMP, B108). Tubes were spun at 35,000 rpm for 2 hr in an SW-40 rotor (Optima L-90K, Beckman Coulter). Each sample was passed through a gradient fractionator (TriaxTM Flow Cell Manual Gradient Station ip, BIOCOMP) and the A260 profile was monitored. 24 fractions with 500 µl of sample each were collected and immediately processed for RNA extraction.

## RNA extraction from sucrose gradient

Fractions were pooled into monosome, low- and high-polysome-associated mRNAs (*Floor and Doudna, 2016*). RNA was extracted by adding equal volumes of acid phenol/chloroform (pH 4.7, Thermo Fisher Scientific, AM9720) to the sucrose fractions. The mix was vortexed briefly and spun down at 14,000 × $g$ at 4°C for 20 min. Subsequently, the aqueous phase was transferred to a new tube. RNA was precipitated with 300 mM Na acetate, ethanol, and 20 µg Glycoblue (Thermo Fisher Scientific, AM9515). The sample was mixed by inversion and stored at –70°C overnight. The RNA was spun down at 14,000 × $g$ at 4°C for 20 min. The pellet was washed with 75% ethanol, vortexed briefly and pelleted at 14,000 × $g$ at 4°C for 5 min. Subsequently, the pellet was air-dried and resuspended in RNase-free water. The purified RNA was quality controlled and used for RNA-seq library preparation.

## RNA-sequencing for ribosome fractions

RNA sequencing libraries were generated using Smart-seq3 V3 protocol suitable for Illumina sequencing from low input total RNA (*Hagemann-Jensen et al., 2020*) (https://www.protocols.io/view/smart-seq3-protocol-bcq4ivyw). The method uses a template switch oligo which has UMIs and a binding site for Nextera read 1 side. Therefore, the method creates UMI labeled, stranded 5' fragments and unstranded fragments all over the transcript body. The size of libraries was assessed using NGS HS analysis kit and a fragment analyzer system (Agilent). Library concentrations were quantified

with KAPA Kit (Roche) and 50 bp paired-read sequencing was performed using the Illumina NovaSeq S1 platform.

## Analysis of RNA-seq of ribosome fractions

Smart-seq3 read analysis was performed using zUMI v2.9.7 (*Parekh et al., 2018*), providing the expected barcodes file, using STAR 2.7.7a with additional STAR parameters '--limitOutSJcollapsed 50000000 --limitIObufferSize 1500000000 --limitSjdbInsertNsj 2000000 --clip3pAdapterSeq CTGTCTCTTATACACATCT'' and *H. sapiens* Ensembl GRCh38 release 94 as a reference. zUMI generated index UMI counts were used for downstream analyses. Differential gene association analysis between groups was performed using DESeq2 wald tests on the umi count tables after minimal pre-filtering where only protein-coding genes with at least two reads in multiple samples of a condition were retained.

## Acknowledgements

We would like to thank Daniel Matějů and the Knoblich lab members for their feedback on the manuscript. We thank the IMBA stem cell core facility for their service, IMBA/IMP/GMI BioOptics facility for flow cytometry and microscopy services; IMBA/IMP/GMI Bioinformatics for sequencing analysis; the VBCF Sequencing unit for sequencing the IMBA/IMP/GMI protein chemistry core facility for mass spectrometry. We thank the group of Andrea Pauli, especially Katrin Friederike Leesch, for sharing the resources and expertise in polysome profiling.

Work in the Knoblich laboratory is supported by the Austrian Academy of Sciences, the Austrian Science Fund (FWF) (Special Research Programme F7804-B and Stand-Alone grants P 35680 and P 35369), the Austrian Federal Ministry of Education, Science and Research, the City of Vienna, and a European Research Council (ERC) Advanced Grant under the European Union's Horizon 2020 programs (no. 695642 and no. 874769). JS was supported by EMBO long term fellowship (EMBO ALTF 794-2018). This project also received funding from the European Union's Horizon 2020 research and innovation program under the Marie Skłodowska-Curie fellowship agreement 841940 awarded to JS and 897137 to PT. Work in the Mechtler lab is supported by the EPIC-XS, project number 823839, funded by the Horizon 2020 Program of the European Union, by the project LS20-079 of the Vienna Science and Technology Fund and by ERA-CAPS I 3686, P35045-B, P32054 (FB) and P33380 (FB) projects of the Austrian Science Fund.

## Additional information

### Competing interests

Jürgen A Knoblich: JAK is inventor on a patent describing cerebral organoid technology (EP2931879B1, US10407664B2) and co-founder and scientific advisory board member of a:head bio AG. The other authors declare that no competing interests exist.

### Funding

| Funder | Grant reference number | Author |
|---|---|---|
| Austrian Science Fund | Special Research Programme F7804-B and Stand-Alone grants P 35680 and P 35369 | Jürgen A Knoblich |
| Horizon 2020 Framework Programme | European Research Council (ERC) Advanced Grant (no. 695642 and no. 874769) | Jürgen A Knoblich |
| European Molecular Biology Organization | EMBO long term fellowship (EMBO ALTF 794-2018) | Jaydeep Sidhaye |

| Funder | Grant reference number | Author |
| --- | --- | --- |
| Horizon 2020 Framework Programme | Marie Skłodowska-Curie fellowship agreement 841940 | Jaydeep Sidhaye |
| Horizon 2020 Framework Programme | Marie Skłodowska-Curie fellowship agreement 897137 | Philipp Trepte |
| Horizon 2020 Framework Programme | EPIC-XS, Project Number 823839 | Karl Mechtler |
| Vienna Science and Technology Fund | project LS20-079 | Karl Mechtler |
| Austrian Science Fund | ERA-CAPS I 3686 | Karl Mechtler |
| Austrian Science Fund | P33380 (FB) | Karl Mechtler |
| Austrian Science Fund | P32054 (FB) | Karl Mechtler |
| Austrian Science Fund | P35045-B | Karl Mechtler |
| Austrian Academy of Sciences | | Jürgen A Knoblich |
| Austrian Federal Ministry of Education, Science and Research | | Jürgen A Knoblich |

The funders had no role in study design, data collection and interpretation, or the decision to submit the work for publication.

## Author contributions

Jaydeep Sidhaye, Conceptualization, Data curation, Funding acquisition, Validation, Investigation, Visualization, Methodology, Writing - original draft, Writing – review and editing; Philipp Trepte, Conceptualization, Data curation, Software, Funding acquisition, Validation, Investigation, Visualization, Methodology, Writing - original draft, Writing – review and editing; Natalie Sepke, Validation, Investigation, Visualization; Maria Novatchkova, Gerhard Dürnberger, Data curation, Software; Michael Schutzbier, Investigation; Karl Mechtler, Resources; Jürgen A Knoblich, Conceptualization, Resources, Supervision, Funding acquisition, Writing – review and editing

### Author ORCIDs

Jaydeep Sidhaye http://orcid.org/0000-0001-7858-8105
Philipp Trepte http://orcid.org/0000-0002-8141-6272
Michael Schutzbier http://orcid.org/0000-0003-4856-262X
Jürgen A Knoblich http://orcid.org/0000-0002-6751-3404

### Decision letter and Author response

Decision letter https://doi.org/10.7554/eLife.85135.sa1
Author response https://doi.org/10.7554/eLife.85135.sa2

## Additional files

### Supplementary files

• Supplementary file 1. List of Oligos used for PCRs, guide sequence cloning and RNA-FISH.

• Supplementary file 2. Summary of genomic integrity and karyotype testing of the cell lines and clones used in this study.

• Supplementary file 3. Reagents and media used in the study.

• MDAR checklist

### Data availability

The RNA-seq data discussed in this publication have been deposited in NCBI's Gene Expression Omnibus (*Edgar et al., 2002*) and are accessible through GEO Series accession numbers GSE214654

(https://www.ncbi.nlm.nih.gov/geo/query/acc.cgi?acc=GSE214654) and GSE214652 (https://www.ncbi.nlm.nih.gov/geo/query/acc.cgi?acc=GSE214652). The mass spectrometry proteomics data have been deposited to the ProteomeXchange Consortium via the PRIDE (*Perez-Riverol et al., 2021*) partner repository with the dataset identifier PXD037106 (https://www.ebi.ac.uk/pride/archive/projects/PXD037106).

The following datasets were generated:

| Author(s) | Year | Dataset title | Dataset URL | Database and Identifier |
|---|---|---|---|---|
| Sidhaye J, Knoblich JA | 2022 | RNA-seq of progenitors and neurons sorted from brain organoids | https://www.ncbi.nlm.nih.gov/geo/query/acc.cgi?acc=GSE214654 | NCBI Gene Expression Omnibus, GSE214654 |
| Sidhaye J, Knoblich JA | 2022 | Results of RNA-seq of ribosome fractions from control and TSC2 KO Day37 organoids | https://www.ncbi.nlm.nih.gov/geo/query/acc.cgi?acc=GSE214652 | NCBI Gene Expression Omnibus, GSE214652 |
| Trepte P, Knoblich JA | 2022 | Transcriptome and proteome analysis of human brain organoids | https://www.ebi.ac.uk/pride/archive/projects/PXD037106 | PRIDE, PXD037106 |

The following previously published dataset was used:

| Author(s) | Year | Dataset title | Dataset URL | Database and Identifier |
|---|---|---|---|---|
| Blair JD, Hockemeyer D, Doudna JA, Bateup HS, Floor SN | 2017 | Widespread translational remodeling during human neuronal differentiation | https://www.ncbi.nlm.nih.gov/geo/query/acc.cgi?acc=GSE100007 | NCBI Gene Expression Omnibus, GSE100007 |

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

# Appendix 1

## Appendix 1—key resources table

| Reagent type (species) or resource | Designation | Source or reference | Identifiers | Additional information |
|---|---|---|---|---|
| Antibody | Chicken anti-GFP, polyclonal | Aves Labs | Aves Labs Cat# GFP-1020; RRID:AB_10000240 | 1:500 |
| Antibody | Goat anti-SOX1, polyclonal | R&D Systems | R&D Systems Cat# AF3369; RRID:AB_2239879 | 1:200 |
| Antibody | Goat anti-Sox2, polyclonal | R&D Systems | R&D Systems Cat# AF2018; RRID:AB_355110 | 1:200 |
| Antibody | Mouse anti-G3BP1, monoclonal | Abcam | Abcam Cat# ab56574; RRID:AB_941699 | 1:200 |
| Antibody | Mouse anti-NeuN, monoclonal | Millipore | Millipore Cat# MAB377; RRID:AB_2298772 | 1:600 |
| Antibody | Rabbit anti-dsred, polyclonal | Takara Bio | Takara Bio Cat# 632496; RRID:AB_10013483 | 1:250 |
| Antibody | Rabbit anti-p4EBP1 (Thr37/46), monoclonal | Cell Signaling Technology | Cell Signaling Technology Cat# 2855; RRID:AB_560835 | 1:200 |
| Antibody | Rabbit anti-pS6 (Ser235/236), monoclonal | Cell Signaling Technology | Cell Signaling Technology Cat# 4858; RRID:AB_916156 | 1:200 |
| Antibody | Rabbit anti-RPL11, polyclonal | Abcam | Abcam Cat# ab79352; RRID:AB_2042832 | 1:200 |
| Antibody | Rabbit anti-RPL5, polyclonal | Abcam | Abcam Cat#ab137617; RRID:AB_2924679 | 1:200 |
| Antibody | Rabbit anti-phospho-EIF2a (Ser51) D9G8, monoclonal | Cell Signaling Technology | Cell Signaling Technology Cat#3398; RRID:AB_2096481 | 1:200 |
| Antibody | Rabbit anti-S100B, monoclonal | Abcam | Abcam Cat# ab52642; RRID:AB_882426 | 1:200 |
| Antibody | Sheep anti-Human EOMES, polyclonal | R&D Systems | R&D Systems Cat#AF6166-SP; RRID:AB_10569705 | 1:200 |
| Antibody | Alexa Fluor 488 Donkey anti-chicken, polyclonal | Jackson ImmunoResearch | Jackson ImmunoResearch Labs Cat# 703-545-155; RRID:AB_2340375 | 1:500 |
| Antibody | Alexa Fluor 488 Donkey anti-goat, polyclonal | Thermo Fisher Scientific | Thermo Fisher Scientific Cat# A-11055; RRID:AB_2534102 | 1:500 |
| Antibody | Alexa Fluor 568 Donkey anti-rabbit, polyclonal | Thermo Fisher Scientific | Thermo Fisher Scientific Cat# A10042; RRID:AB_2534017 | 1:500 |
| Antibody | Alexa Fluor 647 Donkey anti-goat, polyclonal | Thermo Fisher Scientific | Thermo Fisher Scientific Cat# A-21447; RRID:AB_2535864 | 1:500 |
| Antibody | Alexa Fluor 647 Donkey anti-mouse, polyclonal | Thermo Fisher Scientific | Thermo Fisher Scientific Cat# A-31571; RRID:AB_162542 | 1:500 |

*Appendix 1 Continued on next page*

*Appendix 1 Continued*

| Reagent type (species) or resource | Designation | Source or reference | Identifiers | Additional information |
|---|---|---|---|---|
| Antibody | Alexa Fluor 647 Donkey anti-rabbit, polyclonal | Thermo Fisher Scientific | Thermo Fisher Scientific Cat# A-31573; RRID:AB_2536183 | 1:500 |
| Antibody | Alexa Fluor 647 Donkey anti-sheep, polyclonal | Jackson ImmunoResearch | Jackson ImmunoResearch Labs Cat#713-605-147; RRID:AB_2340751 | 1:500 |
| Antibody | Alexa Fluor 488Mouse anti-β-Tubulin, Class III, monoclonal | BD Biosciences | BD Biosciences Cat# 560338; RRID:AB_1645345 | 1:25 |
| Antibody | Alexa Fluor488Mouse IgG2a, κ Isotype control, monoclonal | BD Biosciences | BD Biosciences Cat 558055; RRID:AB_1645612 | 1:25 |
| Antibody | PE Mouse IgG1, κ Isotype Control, monoclonal | BD Biosciences | BD Biosciences Cat# 554680; RRID:AB_395506 | 1:25 |
| Antibody | Sox2 Mouse, PE, Clone: O30-678, BD, monoclonal | BD Biosciences | BD Biosciences Cat 562195; RRID:AB_10895118 | 1:25 |
| Recombinant DNA reagent | pSpCas9(BB)–2A-GFP (PX458) | Addgene | 48138 | |
| Recombinant DNA reagent | AAVS1-Neo-TRE-CMV-Cre-rtTA | Addgene | 165457 | |
| Recombinant DNA reagent | pSF4 TetCMV 5'TOP intron 20xGCN4 Renilla FKBP Stop 24xMS2v5 SV40 CTE polyA | Addgene | 119946 | |
| Recombinant DNA reagent | HR_G3BP1-V5-APEX2-GFP | Addgene | 105284 | |
| Recombinant DNA reagent | pmScarlet-i_C1 | Addgene | 85044 | |
| Commercial assay or kit | Cellartis DEF-CS 500 Culture System | Takara Bio | Y30012 | |
| Commercial assay or kit | Phusion Hot Start Flex DNA Polymerase | New England Biolabs | M0535S | |
| Commercial assay or kit | Human Stem Cell Nucleofector Kit 1 | Lonza | VPH-5012 | |
| Commercial assay or kit | QuickExtract DNA Extraction Solution | Cambio | QE09050 | |
| Commercial assay or kit | NEBNext Ultra II Directional RNA Library Prep Kit for Illumina | New England Biolabs | E7760 | |
| Commercial assay or kit | NEBNext Poly(A) mRNA Magnetic Isolation Module | New England Biolabs | E7490 | |
| Commercial assay or kit | Amaxa nucleofector 2b device | Lonza | AAB-1001 | |
| Other | gentleMACS Dissociator | Miltenyi Biotec | 130-093-235 | see section on 'Organoid dissociation' in the 'Materials and methods' for details |
| Other | FACS ARIAIII | BD Biosciences | | see section on 'FACS' in the 'Materials and methods' for details |
| Other | BD LSR Fortessa Cell Analyzer | BD Biosciences | | see section on 'Flow cytometry analysis' in the 'Materials and methods' for details |
| Other | Epredia Cryostar NX70 cryostat | Thermo Fisher Scientific | 957000H | see section on 'Immunohistochemistry' in the 'Materials and methods' for details |
| Other | Fragment analyser | Agilent | | see section on 'RNA extraction, library generation and RNA-seq' in the 'Materials and methods' for details |
| Other | Gradient master base unit | Biocomp | B108 | see section on 'Polysome profiling by sucrose gradient fractionation' in the 'Materials and methods' for details |

*Appendix 1 Continued*

| Reagent type (species) or resource | Designation | Source or reference | Identifiers | Additional information |
|---|---|---|---|---|
| Software, algorithm | Fiji | https://doi.org/10.1038/nmeth.2019; https://imagej.net/software/fiji/ | | |
| Software, algorithm | trim-galore v0.5.0 | https://www.bioinformatics.babraham.ac.uk/projects/trim_galore/ | | |
| Software, algorithm | bowtie2 v2.3.4.1 | https://bowtie-bio.sourceforge.net/bowtie2/index.shtml | | |
| Software, algorithm | star v2.6.0c | doi:10.1093/bioinformatics/bts635 | | |
| Software, algorithm | subread v1.6.2 | doi:10.1093/nar/gkz114 | | |
| Software, algorithm | DESeq2 v1.18.1 | doi:10.1186/s13059-014-0550-8 | | |
| Software, algorithm | Seurat package | doi:10.1016/j.cell.2021.04.048 | | |
| Software, algorithm | Proteome Discoverer (version 2.4.0.305) | Thermo Fisher Scientific | | |
| Software, algorithm | MSAmanda v2.0.0.13248 | doi:10.1021/pr500202e | | |
| Software, algorithm | maSigPro version 1.68.00 | doi:10.1093/bioinformatics/btu333 | | |
| Software, algorithm | clusterProfiler | doi:10.1089/omi.2011.0118 | | |
| Software, algorithm | biomart | doi:10.1038/nprot.2009.97 | | |
| Software, algorithm | ViennaRNA package | doi:10.1186/1748-7188-6-26 | | |
| Software, algorithm | Transite | doi:10.1016/j.celrep.2020.108064; https://transite.mit.edu/ | | |
| Software, algorithm | REMBRANDTS | https://github.com/csglab/REMBRANDTS; *Alkallas et al., 2017* | | |
| Software, algorithm | zUMI v2.9.7 | doi:10.1093/gigascience/giy059 | | |
| Chemical compound, drug | DAPI | Merck | D9542 | |
| Chemical compound, drug | Doxycycline hyclate | Merck | D9891 | |
| Chemical compound, drug | Sodium (meta)arsenite | Merck | S7400 | |
| Chemical compound, drug | Corning Matrigel hESC-Qualified Matrix | Corning | 354277 | |
| Chemical compound, drug | Puromycin dihydrochloride | Thermo Fisher Scientific | A1113803 | |
| Chemical compound, drug | Geneticin/G418 Sulfate | Thermo Fisher Scientific | 10131035 | |
| Chemical compound, drug | InSolution GSK-3 Inhibitor XVI, CHIR99021 | Merck | 361571 | |
| Chemical compound, drug | Everolimus | Abcam | ab142151 | |
| Chemical compound, drug | DMEM/F12 HEPES | Thermo Fisher Scientific | 11330–032 | |

*Appendix 1 Continued on next page*

*Appendix 1 Continued*

| Reagent type (species) or resource | Designation | Source or reference | Identifiers | Additional information |
|---|---|---|---|---|
| Chemical compound, drug | BSA | Europa Bioproducts | EQBAH-0500 | |
| Chemical compound, drug | 7.5% Sodium bicarbonate | Thermo Fisher Scientific | 25080094 | |
| Chemical compound, drug | Insulin-Transferrin-Selenium (100×) | Thermo Fisher Scientific | 41400–045 | |
| Chemical compound, drug | TGF-beta1 | R&D Systems | RD-240-B-010 | |
| Chemical compound, drug | Accutase | Merck | A6964 | |
| Chemical compound, drug | Anti-adherence rinsing solution | Stemcell Technologies | 7010 | |
| Chemical compound, drug | 10× Trypsin-EDTA | Thermo Fisher Scientific | 15400054 | |
| Chemical compound, drug | TURBO DNase (2 U/µl) | Thermo Fisher Scientific | AM2238 | |
| Chemical compound, drug | DPBS, no calcium, no magnesium (dPBS -/-) | Thermo Fisher Scientific | 14190250 | |
| Chemical compound, drug | Saponin | Merck | 47036 | |
| Chemical compound, drug | Sucrose | Merck | 84097 | |
| Chemical compound, drug | Tissue-Tek O.C.T. Compound | Sakura | 4583 | |
| Chemical compound, drug | Triton X-100 | Merck | 93420 | |
| Chemical compound, drug | Dextran Sulfate 50% Solution | Merck | S4030 | |
| Chemical compound, drug | tRNA from *E. coli* MRE 600 | Merck | 10109541001 | |
| Chemical compound, drug | Ribonucleoside Vanadyl Complex | New England Biolabs | S1402S | |
| Chemical compound, drug | Formamid | Merck | F9037 | |
| Chemical compound, drug | DCX probes | Stellaris | VSMF-2504–5 | |
| Chemical compound, drug | Stellaris RNA FISH wash buffer A | Stellaris | SMF-WA1-60 | |
| Chemical compound, drug | Stellaris RNA FISH wash buffer B | Stellaris | SMF-WB1-20 | |
| Chemical compound, drug | Dako Fluorescence mounting medium | Agilent | S302380-2 | |
| Chemical compound, drug | RNasin Plus Ribonuclease Inhibitor | Promega | NA2615 | |

*Appendix 1 Continued*

| Reagent type (species) or resource | Designation | Source or reference | Identifiers | Additional information |
|---|---|---|---|---|
| Chemical compound, drug | Acid-Phenol:Chloroform, pH 4.5 (with IAA, 125:24:1) | Thermo Fisher Scientific | AM9720 | |
| Chemical compound, drug | Glycoblue | Thermo Fisher Scientific | AM9515 | |
| Cell line (human) | Human embryonic stemm cell (hESC), H9, female | WiCell | WA09 | |
| Cell line (human) | Human induced pluripotent stem cell (hiPSC), Rozh-5, female | hPSCreg | WTSIi015-A | |
| Sequence-based reagent | Sox2_sgRNA-top | This paper | JS89 | CACCgGAGCGGCC CGGTGCCCGGCA |
| Sequence-based reagent | Sox2_sgRNA-bottom | This paper | JS90 | AAACTGCCGGGCAC CGGGCCGCTCc |
| Sequence-based reagent | Sox2_HAL_F | This paper | JS91 | gacggtatcgataagcttgatatcgtc gacCATGATGGAGACGGAGCTG |
| Sequence-based reagent | Sox2_HAL_R | This paper | JS92 | tccgcttccgtcgacCATGTG TGAGAGGGGCAG |
| Sequence-based reagent | P2A-EGFP_F | This paper | JS93 | cccctctcacacatgGTCGAC GGAAGCGGAGCTAC |
| Sequence-based reagent | P2A-EGFP_R | This paper | JS94 | ttcgctgtccggcccTTACTTG TACAGCTCGTCCATGC |
| Sequence-based reagent | Sox2_HAR_F | This paper | JS95 | gagctgtacaagtaaGGGCC GGACAGCGAACTG |
| Sequence-based reagent | Sox2_HAR_R | This paper | JS96 | tggagctccaccgcggtggcgggtttaaac GCAGACTGATTCAAATAATACAGAGCCG |
| Sequence-based reagent | F2DTA_F | This paper | JS56 | GTTTAAACCCGCCACCGC |
| Sequence-based reagent | F2DTA_R | This paper | JS57 | GTCGACGATATCAAGCTTATC |
| Sequence-based reagent | Sox2_PAMmut_F | This paper | JS104 | CCCGGTGCCCGGCA CAGCCATTAACGGCAC |
| Sequence-based reagent | Sox2_PAMmut_R | This paper | JS105 | GTGCCGTTAATGGCTG TGCCGGGCACCGGG |
| Sequence-based reagent | hSyn1_Gibson forward | This paper | OJAB602 | GAAAGAGAGATTTAGAATGA CAGTCTAGAGCGGATGCAT atcgatctgcagagggccctgcgtatg |
| Sequence-based reagent | hSyn1_Gibson Reverse | This paper | OJAB603 | gtcgtgctgagagcgcagccttaagc tgcagaagttggtcgtgaggc actgggcaggtaagtatc |
| Sequence-based reagent | min_CMVpromoter_F | This paper | JS304 | ggtaggcgtgtacggtgg |
| Sequence-based reagent | 5'TOPreportergibson_R | This paper | JS305 | TCCTTAATCAGCTCGCT catggtggctagcctatagtg |
| Sequence-based reagent | gibsontagBFP_F | This paper | JS306 | TAGGCTAGCCACCATG agcgagctgattaaggag |
| Sequence-based reagent | gibsontagBFP_R | This paper | JS307 | CACTGGACTAGTGGATC CGAGCTCGGTACCTCA attaagcttgtgccccag |
| Sequence-based reagent | TSC2_sgRNA-top | This paper | JS376 | CACCGCTTTAGG GCGAGCGTTTGG |
| Sequence-based reagent | TSC2_sgRNA-bottom | This paper | JS377 | AAACCCAAACGC TCGCCCTAAAGC |
| Sequence-based reagent | TSC2_exon5_FP | This paper | JS378 | AGTGGAAGCACTCTGGAAGG |
| Sequence-based reagent | TSC2_exon5_RP | This paper | JS379 | GACGCCGAATCTACATCTCC |
| Sequence-based reagent | TSC2_exon5_seq | This paper | JS380 | CTGCCCTGTACAATGCTGATG |

*Appendix 1 Continued on next page*

*Appendix 1 Continued*

| Reagent type (species) or resource | Designation | Source or reference | Identifiers | Additional information |
|---|---|---|---|---|
| Sequence-based reagent | sox2_F | This paper | JS127 | CAGCTCGCAGACCTACATG |
| Sequence-based reagent | sox2_R | This paper | JS128 | GCACATGATGCTGGACTAG |
| Sequence-based reagent | AAVS1_F | This paper | JAB405 | tgagtccggaccactttgag |
| Sequence-based reagent | hSyn1 promoter_R | This paper | OW23 | ccgcctcatcctggtcc |
| Sequence-based reagent | WPRE_F | This paper | JS125 | gacgtccttctgctacgtc |
| Sequence-based reagent | AAVS1_R | This paper | JAB406 | cttcttggccacgtaacctg |
| Sequence-based reagent | G3BP1_genomic_F | This paper | KNO-oPT-102 | CACTCATTAGTGTTGTGACCC |
| Sequence-based reagent | APEX-N-R | This paper | KNO-oPT-103 | CTCACAGTTGGGTAAGACTTTC |
| Sequence-based reagent | Sox2 guide B | This paper | | GAGCGGCCCGGTGCCCGGCA |
| Sequence-based reagent | G3BP1 guide | This paper | | TCCATGAAGATTCACTGCCG |
| Sequence-based reagent | TSC2 guide | This paper | | CTTTAGGGCGAGCGTTTGG |
| Sequence-based reagent | RPL5_1 | Stellaris, Quasar 570 Dye | | cgctagggggtgggaaaagg |
| Sequence-based reagent | RPL5_2 | Stellaris, Quasar 570 Dye | | catcctgcggaacagagacc |
| Sequence-based reagent | RPL5_3 | Stellaris, Quasar 570 Dye | | gccttattcttaacaacttt |
| Sequence-based reagent | RPL5_4 | Stellaris, Quasar 570 Dye | | cacttggtatctcttaaagt |
| Sequence-based reagent | RPL5_5 | Stellaris, Quasar 570 Dye | | cctctcgtcgtcttctaaat |
| Sequence-based reagent | RPL5_6 | Stellaris, Quasar 570 Dye | | tccgagcataataatcagtt |
| Sequence-based reagent | RPL5_7 | Stellaris, Quasar 570 Dye | | ttatcttgtatcaccaagcg |
| Sequence-based reagent | RPL5_8 | Stellaris, Quasar 570 Dye | | ctgtatttgggtgtgttgta |
| Sequence-based reagent | RPL5_9 | Stellaris, Quasar 570 Dye | | ctctgtttgtcacacgaact |
| Sequence-based reagent | RPL5_10 | Stellaris, Quasar 570 Dye | | acgggcataagcaatctgac |
| Sequence-based reagent | RPL5_11 | Stellaris, Quasar 570 Dye | | gctgcgcagactatcatatc |
| Sequence-based reagent | RPL5_12 | Stellaris, Quasar 570 Dye | | acaccatattttggcagttc |
| Sequence-based reagent | RPL5_13 | Stellaris, Quasar 570 Dye | | agcataatttgtcaggccaa |
| Sequence-based reagent | RPL5_14 | Stellaris, Quasar 570 Dye | | cagcaggccagtacaatatg |
| Sequence-based reagent | RPL5_15 | Stellaris, Quasar 570 Dye | | ccaaacctattgagaagcct |
| Sequence-based reagent | RPL5_16 | Stellaris, Quasar 570 Dye | | cttggccttcatagatcttg |
| Sequence-based reagent | RPL5_17 | Stellaris, Quasar 570 Dye | | ttgtattcatcaccagtcac |
| Sequence-based reagent | RPL5_18 | Stellaris, Quasar 570 Dye | | tggctgaccatcaatgcttt |

*Appendix 1 Continued on next page*

Appendix 1 Continued

| Reagent type (species) or resource | Designation | Source or reference | Identifiers | Additional information |
|---|---|---|---|---|
| Sequence-based reagent | RPL5_19 | Stellaris, Quasar 570 Dye | | tccaaatagcaggtgaaggc |
| Sequence-based reagent | RPL5_20 | Stellaris, Quasar 570 Dye | | cagtggtagttctggcaagg |
| Sequence-based reagent | RPL5_21 | Stellaris, Quasar 570 Dye | | cttcagggcaccaaaaactt |
| Sequence-based reagent | RPL5_22 | Stellaris, Quasar 570 Dye | | tgtgagggatagacaagcct |
| Sequence-based reagent | RPL5_23 | Stellaris, Quasar 570 Dye | | taaccagggaatcgtttggt |
| Sequence-based reagent | RPL5_24 | Stellaris, Quasar 570 Dye | | ctgcattaaattccttgctt |
| Sequence-based reagent | RPL5_25 | Stellaris, Quasar 570 Dye | | atgatgtgcttccgatgtac |
| Sequence-based reagent | RPL5_26 | Stellaris, Quasar 570 Dye | | gtaatctgcaacattctggc |
| Sequence-based reagent | RPL5_27 | Stellaris, Quasar 570 Dye | | tcttcttccattaagtagcg |
| Sequence-based reagent | RPL5_28 | Stellaris, Quasar 570 Dye | | actgtttcttgtaagcatct |
| Sequence-based reagent | RPL5_29 | Stellaris, Quasar 570 Dye | | ggagttacgctgttctttat |
| Sequence-based reagent | RPL5_30 | Stellaris, Quasar 570 Dye | | gagctttcttatacatctcc |
| Sequence-based reagent | RPL5_31 | Stellaris, Quasar 570 Dye | | tggattctctcgtatagcag |
| Sequence-based reagent | RPL5_32 | Stellaris, Quasar 570 Dye | | tcttgggcttcttttcatag |
| Sequence-based reagent | RPL5_33 | Stellaris, Quasar 570 Dye | | ccacctcttcttttttaactt |
| Sequence-based reagent | RPL5_34 | Stellaris, Quasar 570 Dye | | tgagcaagggacattttggg |
| Sequence-based reagent | RPL5_35 | Stellaris, Quasar 570 Dye | | ttcttttgagctacccgatc |
| Sequence-based reagent | RPL5_36 | Stellaris, Quasar 570 Dye | | ctgagctctgaggaagcttg |
| Sequence-based reagent | RPL5_37 | Stellaris, Quasar 570 Dye | | aaattgctgggtttagctct |
| Sequence-based reagent | RPL5_38 | Stellaris, Quasar 570 Dye | | agttgctgttcataagttta |
| Sequence-based reagent | RPL11_1 | Stellaris, Quasar 570 Dye | | ccatgatggagagcaggaag |
| Sequence-based reagent | RPL11_2 | Stellaris, Quasar 570 Dye | | ttctcctttcaccttgatc |
| Sequence-based reagent | RPL11_3 | Stellaris, Quasar 570 Dye | | agtttgcggatgcgaagttc |
| Sequence-based reagent | RPL11_4 | Stellaris, Quasar 570 Dye | | cccaacacagatgttgagac |
| Sequence-based reagent | RPL11_5 | Stellaris, Quasar 570 Dye | | tggaaaacacaggggtctgc |
| Sequence-based reagent | RPL11_6 | Stellaris, Quasar 570 Dye | | gatctgacagtgtatctagc |
| Sequence-based reagent | RPL11_7 | Stellaris, Quasar 570 Dye | | atcttttcatttctccggat |
| Sequence-based reagent | RPL11_8 | Stellaris, Quasar 570 Dye | | tcgaactgtgcagtggacag |
| Sequence-based reagent | RPL11_9 | Stellaris, Quasar 570 Dye | | ttctccaagatttcttctgc |

Appendix 1 Continued on next page

*Appendix 1 Continued*

| Reagent type (species) or resource | Designation | Source or reference | Identifiers | Additional information |
|---|---|---|---|---|
| Sequence-based reagent | RPL11_10 | Stellaris, Quasar 570 Dye | | tttcttaactcatactcccg |
| Sequence-based reagent | RPL11_11 | Stellaris, Quasar 570 Dye | | tccagtatctgagaagttgt |
| Sequence-based reagent | RPL11_12 | Stellaris, Quasar 570 Dye | | cctggatcccaaaaccaaag |
| Sequence-based reagent | RPL11_13 | Stellaris, Quasar 570 Dye | | tttgatacccagatcgatgt |
| Sequence-based reagent | RPL11_14 | Stellaris, Quasar 570 Dye | | tagataccaatgcttgggtc |
| Sequence-based reagent | RPL11_15 | Stellaris, Quasar 570 Dye | | agcaccacatagaagtccag |
| Sequence-based reagent | RPL11_16 | Stellaris, Quasar 570 Dye | | tcttgtctgcgatgctgaaa |
| Sequence-based reagent | RPL11_17 | Stellaris, Quasar 570 Dye | | tctttgctgattctgtgttt |
| Sequence-based reagent | RPL11_18 | Stellaris, Quasar 570 Dye | | atgatcccatcatacttctg |
| Sequence-based reagent | RPL11_19 | Stellaris, Quasar 570 Dye | | acgggaatttatttgccagg |
| Sequence-based reagent | RPL11_20 | Stellaris, Quasar 570 Dye | | ctttttattgctctttttgga |
| Recombinant DNA reagent | pSpCas9(BB)–2A-tomato SOX2-sgRNA | This study; Addgene | 196190 | |
| Recombinant DNA reagent | pHDR_sox2HA_P2A-GFP-pA_HA_G2913A | This study; Addgene | 196191 | |
| Recombinant DNA reagent | AAVS1 SA-2A-puro-pA_hSYN1_dTomato-SV40pA | This study; Addgene | 196192 | |
| Recombinant DNA reagent | AAVS1-Neo-TRE-CMV-5'TOPtagBFP_bGHpA-CAG rtTA | This study; Addgene | 196193 | |
| Recombinant DNA reagent | AAVS1-Neo-TRE-CMV-mutant5'TOPtagBFP_bGHpA-CAG rtTA | This study; Addgene | 196194 | |
| Recombinant DNA reagent | pSpCas9(BB)–2A-tomato-TSC2-sgRNA | This study; Addgene | 196195 | |
| Recombinant DNA reagent | pHDR_G3BP1-V5-APEX2-mScarlet-I EF1a-Puro | This study; Addgene | 196196 | |
| Recombinant DNA reagent | pSpCas9(BB)–2A-Puro G3BP1-sgRNA | This study; Addgene | 196197 | |

